# Specialised and persistent raw material procurement by humans in the Middle Pleistocene

Manuel Will [1,2] ✉, Christian Sommer [3,4], Gunther H. D. Möller[1], Greg A. Botha[5], Matthias A. Blessing [1,2,6], Lawrence Msimanga[7], Aron Mazel[8], Aurore Val[9,10,11], Flavia Venditti [1] & Svenja Riedesel [12,13]

The selection and acquisition of suitable raw material constitute the first steps in stone tool technology. Previous ethnographical and archaeological research suggests that hominins in the Pleistocene primarily collected their stone materials while carrying out other activities. Direct provisioning for this purpose alone remains an outlier and is rarely demonstrated. Archaeological excavations coupled with multidisciplinary analyses at Jojosi in South Africa demonstrate that early modern humans undertook specific, repeated visits to a raw material source over tens of thousands of years for the exclusive purpose of obtaining hornfels. This rare, stratified, open-air locality features uniquely preserved lithic assemblages with abundant refits dating from ~220 ka to ~110 ka for the reduction and export of a single tool stone. The scope of these knapping activities is underscored by millions of Middle Stone Age hornfels artefacts paving the modern landscape. The consistent, specialised procurement of a single raw material at Jojosi already during the Middle Pleistocene challenges the standard model of embedded procurement for this period. These findings further show that key capacities of *Homo sapiens*, including increased long-term planning and behavioural plasticity in the interaction with the material world, emerged early in their evolutionary history.

Stone tools knapped from a wide range of rock types make up the principal archaeological evidence for the Palaeolithic era since the earliest artefacts at ca. 3.3–2.6 Myra[1,2]. How hominins acquired, reduced, and used various lithic materials remains central for understanding their past behaviour and cultural evolution. Studies of the nature, origin, quality, and transport distances of tool stone throughout the Pleistocene have been key aspects of archaeological scholarship. The topic concerns basic motor capabilities of knapping

[1]Working Group Early Prehistory and Quaternary Ecology, Department of Geosciences, Faculty of Science, University of Tübingen, Tübingen, Germany. [2]Palaeo-Research Institute, University of Johannesburg, Johannesburg, South Africa. [3]University of Tübingen, Department of Geosciences, Institute of Geography, Tübingen, Germany. [4]The Role of Culture in Early Expansions of Humans, Heidelberg Academy of Sciences and Humanities, Tübingen, Germany. [5]Evolutionary Studies Institute, University of the Witwatersrand, Johannesburg, South Africa. [6]University of Connecticut, Department of Anthropology, Deep History Lab, Storrs, CT, USA. [7]ArcheoTask, Engen, Germany. [8]Newcastle University, Department of Media, Culture, Heritage, Newcastle upon Tyne, Great Britain. [9]CNRS, Aix Marseille Université, Ministère de la Culture, LAMPEA, UMR, Aix-en-Provence, France. [10]School of Geography, Archaeology and Environmental Studies, University of the Witwatersrand, Johannesburg, South Africa. [11]ICArEHB - Interdisciplinary Center for Archaeology and the Evolution of Human Behaviour, Universidade do Algarve, Campus de Gambelas, Faro, Portugal. [12]Institute of Geography, University of Cologne, Cologne, Germany. [13]Luminescence Physics and Technologies, Department of Physics, Technical University of Denmark, Roskilde, Denmark. ✉e-mail: manuel.will@uni-tuebingen.de

rocks of various textures as well as wide-ranging interpretations on past mobility patterns, behavioural flexibility, the organisation of technologies, and cognitive capacities such as planning depth[3–16].

Living a mobile, hunter-gatherer lifestyle, Pleistocene hominins had various options to obtain their tool stones, ranging from collecting these resources themselves to exchanging them with other groups. The dominant paradigm on how hominins collected their material stems from influential ethnographic work on recent hunter-gatherers, which distinguishes two principal modes[3–5]. The first and most frequent one is indirect or embedded procurement, according to which individuals obtain their raw materials as they encounter them in the landscape during the execution of other tasks, such as hunting prey or gathering food, as a means of saving time and energy. A second and less commonly documented method consists of direct or specialised procurement, which denotes planned trips undertaken by special task groups to acquire a specific tool stone from a particular location, not directly associated with other activities.

While these seminal ethnographic works were based on historical hunter-gatherers, most archaeological studies have assumed or shown that embedded procurement of various tool stones was the major and standard mode throughout the Palaeolithic (e.g., refs. 6–11). Even long-distance transport of raw material over >50 km, first associated with the Middle Stone Age (MSA, ~300–30 ka), has been commonly interpreted as the result of embedded procurement practices with increased group movement and/or inter-group exchange[10,17–20]. Scholars have considered direct acquisition of materials either as an outlier behaviour or as exceedingly difficult to trace in the deep archaeological past. The best-known, demonstrated cases for direct

provisioning consist of large-scale mining or quarrying activities widely found in the Neolithic[21,22], but such behaviours are rarely known from the Pleistocene. Among the few well-known early cases are (i) the MSA assemblages of Taramsa 1 in Egypt, where extensive chert quarrying is demonstrated starting at ~166 ka[12,13], (ii) whole landscapes for flint extraction from the Middle Palaeolithic of the Near East (though they remain undated due to their surface nature)[23,24], (iii) the mining of ochre in Eswatini at ~40 ka[25], and (iv) specialised Upper Palaeolithic workshops and underground mines after ~40 ka in northern Africa, Europe, and Australia[12,26,27]. Apart from surface occurrences[28,29], no stratified and dated site for comparable extraction and specialised procurement of tool stone is known from the Middle or Late Pleistocene of sub-Saharan Africa. Based on current knowledge, direct acquisition may either not have been part of the standard behavioural repertoire of Pleistocene hominins or a rather late development. This is particularly evident in southern Africa, which has the best-resolved archaeological record of early *Homo sapiens* during the MSA, and a well-represented fossil record including specimens such as Florisbad (~260 ka), Border Cave (~170–80 ka), and Klasies River Mouth (~120–60 ka)[30,31].

Here, we report on the excavation of a unique, stratified open-air site at Jojosi in the understudied grassland region of eastern South Africa, located roughly halfway between Florisbad and Border Cave, that provides important insights into raw material provisioning by early modern humans during the Pleistocene (Fig. 1). Multiple in situ assemblages of stone artefacts occur in a complex erosional landscape (known locally as 'dongas') and are exceptionally well-preserved. They occur adjacent to extensive primary outcrops and secondary deposits

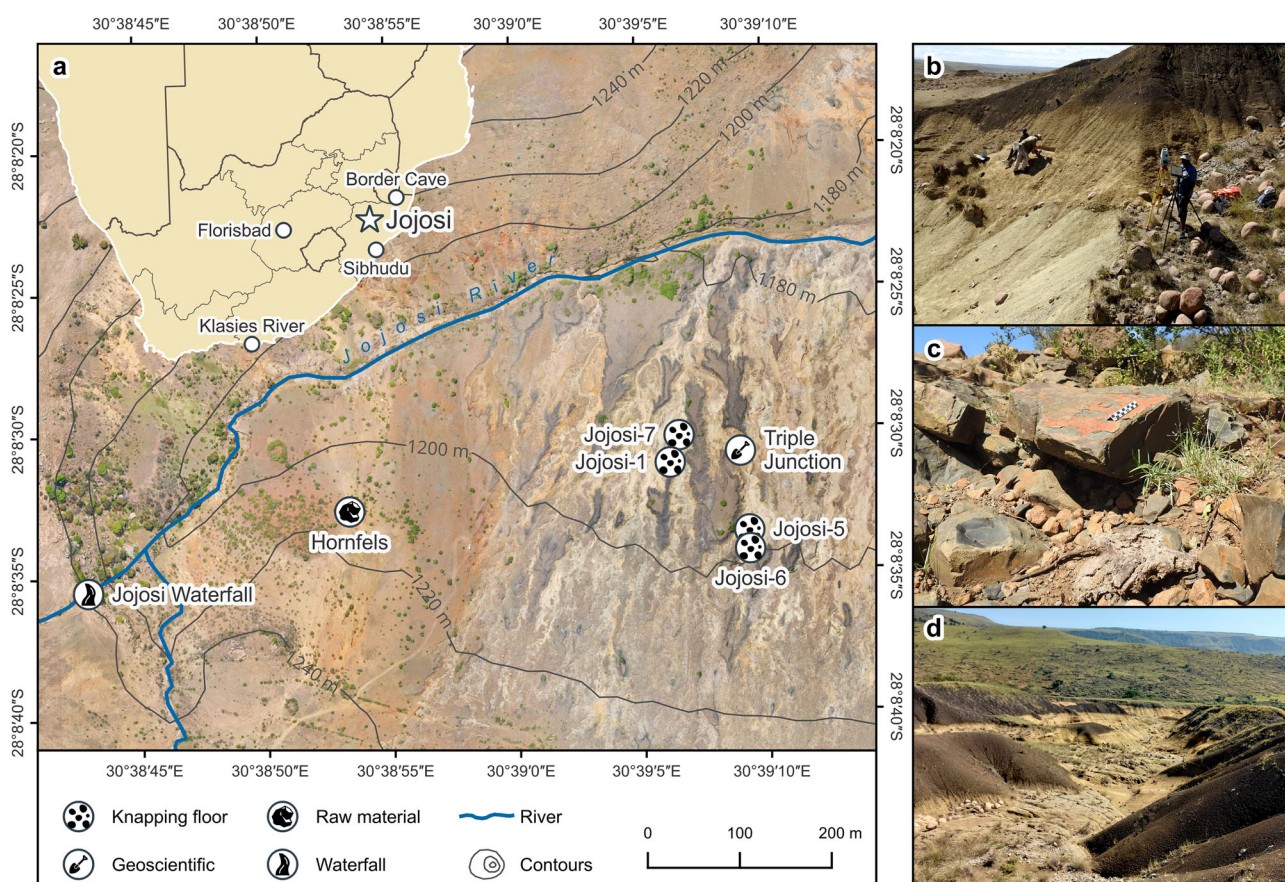

**Fig. 1 | Map of the Jojosi landscape and an inset map of southern Africa with sites that have yielded early *H. sapiens* fossil material. a** View on the excavation of Jojosi 6 in 2023. **b** Primary hornfels outcrop as a raw material source for stone tool knapping. **c** Exemplary view of the erosional landscape of the Jojosi Dongas. **d** Map icons in (**a**) by Dong Ik Seo, Michael Wohlwend, MihiMihi and Muhammad Nur Auliady Pamungkas from Noun Project (CC BY 3.0). Site locations in (**a**) from the ROAD database[101].

of hornfels, a fine-grained metamorphic rock used frequently in the Stone Age of Africa to produce tools. Multidisciplinary analyses of Jojosi's resource landscape and MSA lithic assemblages demonstrate large-scale, targeted raw material procurement from these hornfels sources combined with specialised lithic workshops dedicated to the production and subsequent export of blanks by *Homo sapiens*. These activities started already in the Middle Pleistocene (-220 ka) and endured unchanged for many tens of thousands of years until the Late Pleistocene (-110 ka). The Jojosi findings transform our understanding of how early *Homo sapiens* organised their raw material procurement and have major implications for their capacity for long-term planning, behavioural flexibility, and material engagement.

## Results

### The Jojosi Dongas

The open-air site of Jojosi represents an extensive landscape (-1 km²) of erosional gullies below a dolerite foot slope and above the Jojosi River, where the chemically weathered dolerite bedrock is buried by a complex sequence of Quaternary sheetwash and gully-infill sediments (Fig. 1). The area is situated in the Southeastern Coastal Hinterland of KwaZulu-Natal (KZN), approximately 140 km from the coast, at an elevation of -1200 m. The site lies within a modern subtropical highland climate (Köppen-Geiger class Cwb) where the predominant vegetation type is the Income Sandy Grassland[32]. Pleistocene geomorphological processes at Jojosi created a complex topography of gully channels cutting into foot slope deposits, thereby exposing both sediments and a lag deposit comprising abundant stone artefacts on the modern surface and, in rarer cases, artefact lenses that actively erode from the donga walls (Supplementary Note 1). The area was first studied geologically in the 1980s[33], and the archaeological potential was tested by excavations of four locations (Jojosi 1–4) by A. Mazel in 1991. The results were never published, but the artefact collections have been well-documented and curated. The Southeastern Coastal Hinterland has received much Holocene work[34], but no Pleistocene archaeological research has been undertaken. As a result, no stratified Early Stone Age (ESA) or MSA sites exist within a radius of 150 km around Jojosi[35]. This situation differs from the better-known MSA localities in KZN to the south and closer to the Indian Ocean, in particular Sibhudu Cave[36,37], and others[38,39], and the Western Cape coastlines (e.g., refs. 40,41).

Since 2022, our interdisciplinary team has studied the legacy museum collections and undertaken field and analytical work on the geology and archaeology of the Jojosi landscape and its formation processes[35,42]. Numerous cut-and-fill cycles are represented by the sediment bodies, which indicate episodic phases of geomorphic landscape stability and instability. The stratigraphic sequence reflects the geomorphic processes observed today, such as sheet erosion, channel incision, and gully sidewall collapse, but also deposition under various energetic regimes, ranging from alluvial channel deposits to colluvial deposition (Supplementary Note 1). Our geological surveys documented a landscape with diverse resources, including drinking water from the adjacent river and a nearby waterfall, various accessible rock resources, and several cliffs that provide shade and shelter (Fig. 1). Situated in a topographically diverse landscape with orographic effects on local climate, the area offers various plant and animal resources at the grassland–savanna interface[32]. Palaeoclimatic reconstructions indicate fluctuating conditions for the general area that shifted between grassland and savanna environments, with precipitation levels and seasonality sometimes slightly lower and sometimes higher than today during the last 200,000 years, providing enough rainfall to support at least a seasonal flow of the Jojosi River[43,44]. Most importantly, we identified large primary outcrops of in situ hornfels -500 m distant from the archaeological excavations and partly buried under alluvial sediments, as well as large, angular blocks washed down into the landscape, that are preserved within the alluvial terrace gravel

and the erosional gullies incising the hillslope adjacent to these sources (Fig. 1c; see also Supplementary Fig. 11). The hornfels formed through the baking of the local argillaceous rocks (siltstone) and the physiochemical alteration by contact-metamorphism with the hot igneous magma intrusions that formed the dolerite bedrock. The resulting material is grey to dark grey, fine-grained, and possesses good knapping quality, such as low knapping force requirements[45,46]. Hornfels was commonly used throughout the Stone Age in southern Africa by various hominin species, particularly by modern humans in the MSA and Later Stone Age (LSA)[28,34,37,45].

Our extensive archaeological foot and drone surveys in 2022 and 2023 encountered abundant lithic artefacts in various stages of weathering, covering the irregular gullied surface as an extensive lithic pavement of worked stone[35]. Most often, they occur as lag deposits on the floor of incised gullies but also on interfluve ridges consisting of intact soil-covered remnants of the original surface (Supplementary Figs. 11 and 12). Almost all diagnostic stone artefacts have an MSA character. Rare ESA handaxes occur only in small parts of the landscape, and not in the areas we excavated. Virtually all of the surface tools are made from hornfels. Given the existence of large outcrops of dolerite next to the site, and quartz and quartzite pebbles obtainable in the Jojosi River, this was a surprising observation. During the survey, we discovered about a dozen instances in which stratified, nonweathered hornfels artefacts were actively eroding from the sedimentary profiles. We targeted these occurrences as the highest excavation priority, as potentially providing remaining in situ material and rescuing it before complete erosion.

In 2023 and 2024, we excavated five of these artefact lenses at three different locations within the dongas, named Jojosi 5, 6, and 7. These stratified archaeological assemblages occur exclusively within Unit 4 of the geological stratigraphic sequence—a unit ranging between 1 and 5 m in thickness, characterised by well-sorted and thinly bedded deposits of clay and sandy loam with low content of coarse clasts and hardened through cementation (see Supplementary Note 1)—at various elevations within this sediment body and typically capped by modern topsoil (Fig. 2). Due to the absence of internal stratigraphy of Unit 4, excavations proceeded in contexts to carefully uncover the sediments above, within, and below the artefact lenses (see Methods). To facilitate high-precision recovery and documentation of finds, we plotted all artefacts >2 cm using a Total Station and sieved all sediments down to a 1 mm mesh size. We also analysed the legacy museum assemblages from the 1991 excavations by Mazel at Jojosi 1. Triangulating from available photos, we were able to identify the location of this site, which allowed luminescence dating of its remaining sediments[42,47]. This article presents the first description of the recently excavated archaeological assemblages in combination with the site formation processes and luminescence dating of all stratified locations.

### Site formation and chronology

A central concern of the work at Jojosi has been to demonstrate the in situ character of the stratified artefact assemblages due to their open-air nature, particularly in the context of Pleistocene cut-and-fill geomorphic processes. During excavation, artefact lenses were encountered as thin bands of stone tool clusters with very high concentrations (find densities between 200,000 and 2,000,000 n/m³; Supplementary Table 8), demarcated in space both vertically and horizontally (Fig. 3). The sediment contexts removed above, below, and to the sides of the lenses yielded no or very low amounts of lithic artefacts, with find densities being magnitudes below the lenses (relative densities of lenses compared to overburden range between multiples of -300–1400; Supplementary Table 8; Supplementary Figs. 19–21, 29, 33). Typically, artefacts within lenses lie densely stacked on top of each other (Supplementary Figs. 14, 16 and 27). The size distribution of lithics for the artefact lenses at Jojosi 5 and 6 matches undisturbed

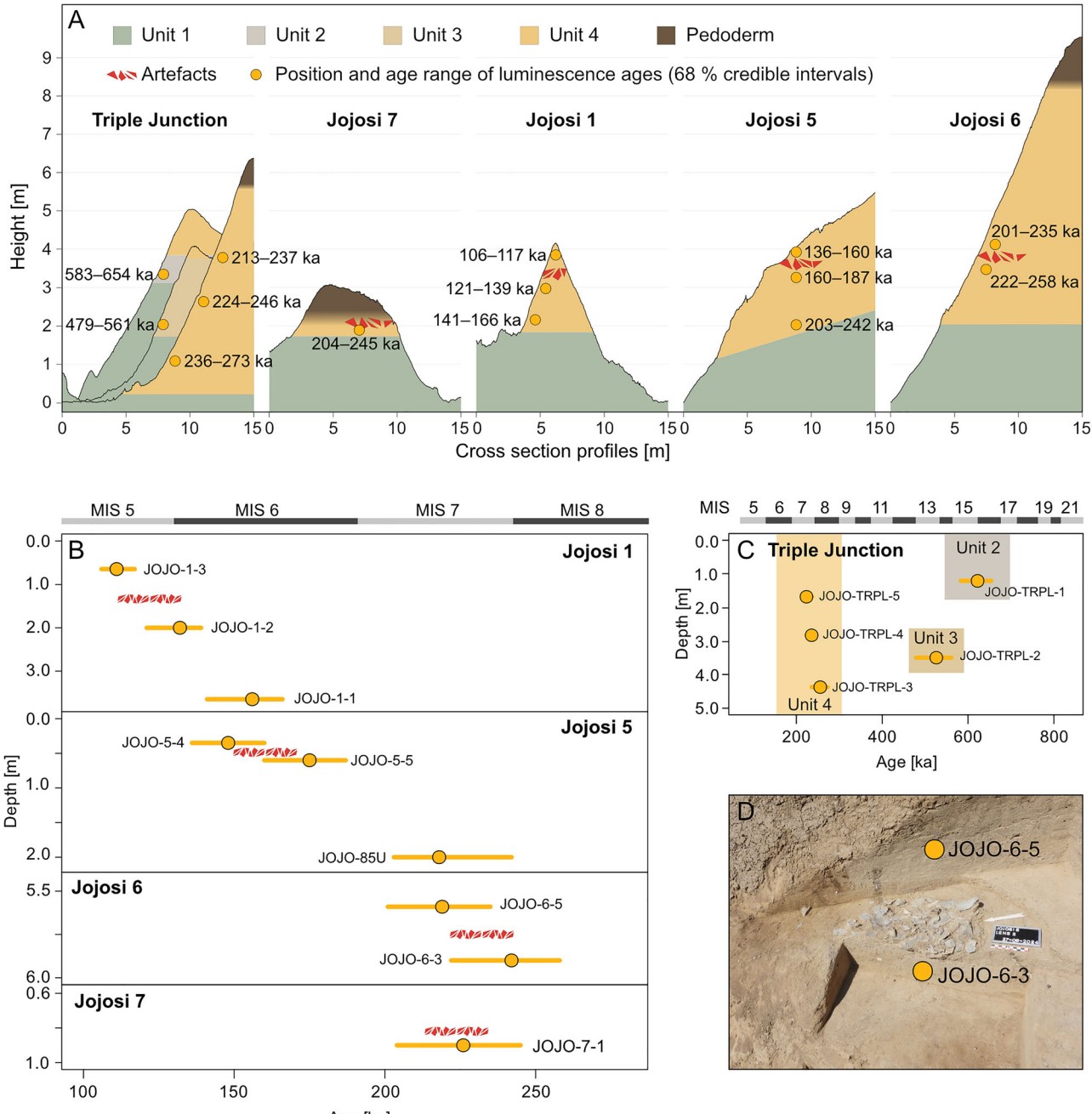

**Fig. 2 | Overview of the stratigraphy and chronometric dating of Jojosi.**
**A** Schematic geological stratigraphy for all sites, including the position of luminescence samples and archaeological remains. The luminescence ages are displayed as mean BayLum[79] ages with their 68% credible interval. Triple Junction is the key geological reference stratigraphy for Jojosi. The archaeological sites Jojosi 1, 5, 6, and 7 all lie within Unit 4. **B** Age depth profiles of the four archaeological sites. The luminescence ages are displayed as the mean BayLum age with their respective 68% credible interval. **C** Age depth file of the key geological reference site, the Triple Junction. The luminescence ages are displayed as their respective 68% credible interval. **D** Photograph depicting the exact relationship between the artefact lens and the luminescence samples at Jojosi 6. Source data are provided in the Source Data file.

experimental knapping workshops of hornfels[48], with very high amounts of microdebris <5 mm and comparatively few pieces >2 cm (Supplementary Table 7). Rose diagrams of artefact orientation measured in the field on elongated pieces illustrate a predominantly random pattern (Supplementary Fig. 41). Geological and chronometric observations confirm sedimentation in a low-energy depositional environment for the artefact-bearing sediments, with conditions typically associated with sheetwash deposits and soil creep: thinly layered and well-sorted clayey to sandy loamy deposits with low inclination (Supplementary Note 1). Use-wear analyses of selected

artefacts at Jojosi 5 and 6 demonstrate the fresh state of the artefacts. These analyses found no heavy patina and no weathering or abrasion of the edges, ridges, or surfaces that would have resulted from long surface exposure or intense transport (Supplementary Note 5).

To further substantiate the apparent in situ preservation, we undertook systematic refitting within all lenses. For Jojosi 1, 5, 6, and 7, we were able to refit a total of 353 artefacts >2 cm in 123 refit groups, amounting to between 5.9 and 49.3% of the individual assemblages with a global average of 15.6% (Supplementary Table 9). The refits at all locations include frequent conjoined breaks, but also refits of

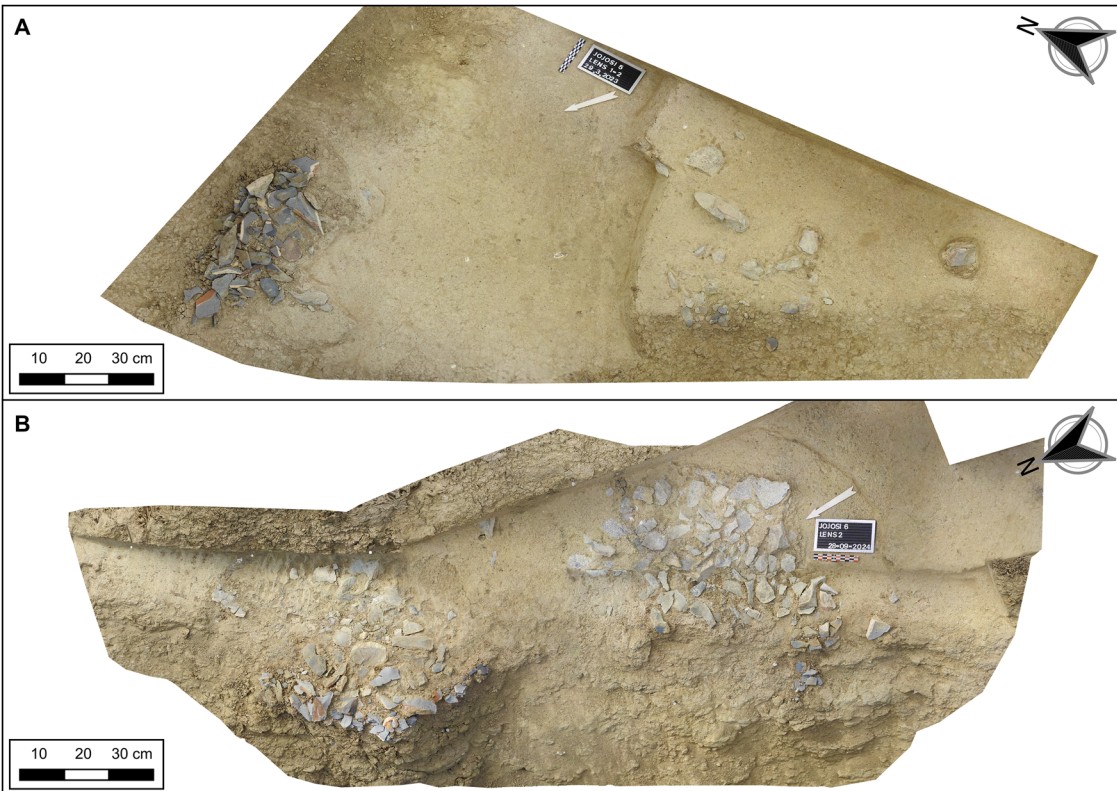

**Fig. 3 | Structure from Motion (SfM) orthophotographic overviews of the excavated artefact lenses and surfaces at Jojosi 5 and 6. A** Jojosi 5 Lenses 1 and 2 excavated in 2023. Scale 1:12. True North follows the measurement grid; the arrow of the excavation documentation in the photo is slightly displaced. **B** Jojosi 6 Lenses 1 and 2 excavated in 2023 and 2024. Scale 1:12. Merged and blended orthophotos from both excavation seasons. Both were created in Agisoft Metashape Professional 2.1.4 and QGIS 3.34. Orthophoto constructed from SfM-3D-Model with overlaid texture and georeferenced to a local coordinate system with artefact measurements.

technological reduction sequences (Fig. 4). The spatial configuration of refits at all lenses supports an unusually high level of stratigraphic integrity and completeness with refit distances of <30 cm (Supplementary Tables 10–14). Most lenses featured exclusively knapped stone artefacts and potential hammer stones. The only outlier is Jojosi 7, which yielded some faunal remains, albeit heavily fragmented and burned. Zooarchaeological analysis suggests that all plotted material ($n = 26$) belongs to a single mandible of a large bovid (size class III) with two pieces showing cut-marks (Supplementary Note 6). Overall, various lines of evidence converge to support the in situ nature of all stratified artefact lenses with remarkably high integrity and minor post-depositional impact (e.g., trampling and sedimentary pressure), resulting from the primary purpose of stone knapping during repeated, short events.

Luminescence dating at both archaeological and geological points of interest (see Methods) provided secure absolute ages for the Jojosi sites. The dating samples from geological locations ($n = 5$) suggest an ancient erosional landscape with successive cycles of deposition and erosion dating from at least the early Middle Pleistocene to the early Late Pleistocene (>600–110 ka; Fig. 2, see ref. 47 and Supplementary Note 2), and a high sedimentation rate at the archaeological sites. We dated a total of nine samples associated with archaeological occurrences (Jojosi 1, $n = 3$; Jojosi 5, $n = 3$; Jojosi 6, $n = 2$; Jojosi 7, $n = 1$) within geological Unit 4 to bracket each of the artefact lenses. The results based on luminescence dating and stratigraphic correlations (Table 1, Fig. 2) demonstrate an age gradient between the locations from oldest to youngest: Jojosi 6 bracketed at ~220 ka (201–258 ka, 1 sigma) with slightly younger ages at Jojosi 7 (<204–245 ka), a later and non-overlapping age for Jojosi 5 at ~160 ka (136–187 ka, 1 sigma) and the latest occurrence at Jojosi 1 at ~110 ka

(106–139 ka), with only a minimal overlap between the ages of the underlying age bracket of Jojosi 5 and the overlying bracket for Jojosi 1. These ages confirm a repeated use of the landscape over multiple phases of short visits that appear to have occurred during many tens of thousands of years, starting well into the Middle Pleistocene (MIS 7 and 6) and extending at least until the early Late Pleistocene (MIS 5e). The apparent frequency of past visits and the extraordinary time depth of these activities match the remarkable number of comparable MSA stone artefacts on the surface, but remain a conservative assessment considering the existence of known additional occupations in the landscape. We identified further stratified lenses of analogous character at Jojosi during previous surveys in various stratigraphic positions within Unit 4 that remain unexcavated, and ongoing donga erosion has likely destroyed many more that once existed, leading to the ubiquitous modern surface record[35]. With these dates, Jojosi 6 joins the lowermost deposits at Border Cave (~227 ± 11 ka)[31] as the oldest known MSA occurrences in KZN.

## Stone tools and raw material provisioning

Jojosi 5, 6, and 7 yielded a combined total of 20,853 lithic artefacts, with only a small fraction being >20 mm ($n = 1443$; 6.9%). Even though the 1991 excavation at Jojosi 1 did not adopt the same methods of systematic recovery of small finds, the assemblage provides a broadly comparable pattern with only 1232 lithics >20 mm (16.4%) and 6297 pieces <20 mm (83.4%). Small flaking debris dominates all excavated assemblages, such as at Jojosi 6 lens 1 and 2, where we recovered 4961 and 3802 pieces of microdebris <5 mm, respectively (Supplementary Table 7). All Jojosi knapped artefacts, regardless of size, are from hornfels. We found no other raw material in the lenses, including Jojosi 1, thereby matching the surface finds.

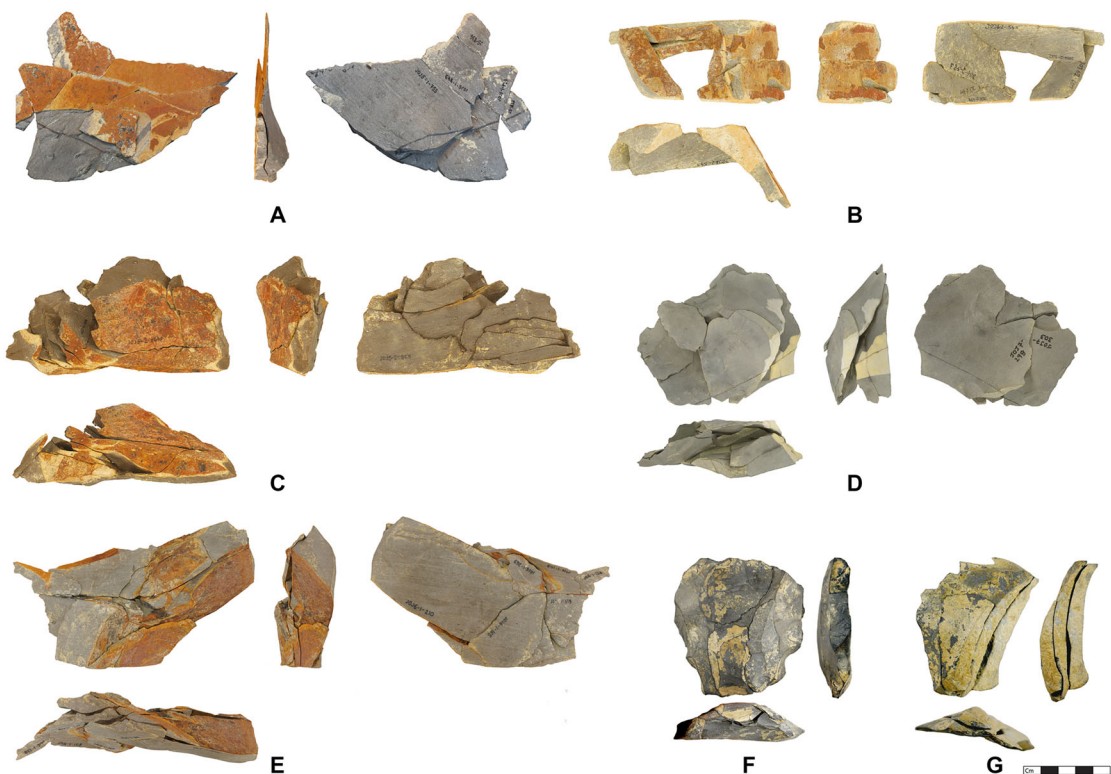

**Fig. 4 | Conjoined and refitted artefacts in refitting groups (RG) from the Jojosi excavations in 1991, 2023, and 2024. A** Jojosi 5 Lens 1, RG 17 with *n* = 9 artefacts with production sequences and breaks in a unidirectional reduction sequence during initial stage decortification. **B** Jojosi 6 Lens 2, RG 100 with *n* = 9 artefacts with production sequences and breaks in an initial stage reduction sequence of a core opening. **C** Jojosi 5 Lens 2, RG 6 with *n* = 19 artefacts with production sequences and breaks in long reduction sequence after prior blade production with multiple reorientations of the core until its discard. **D** Jojosi 7 Lens 1, RG 53 with *n* = 7 artefacts with production sequences in an orthogonal reduction sequence detaching several *débordant* flakes at the lateral core edges. **E** Jojosi 6 Lens 1, RG 38 with *n* = 15 artefacts with production sequences and breaks in a long reduction sequence with multiple reorientations of the core until its discard. **F** Jojosi 1, RG 2 with *n* = 4 artefacts with production sequences displaying the final three operations before discard on an atypical, bidirectionally reduced Levallois core. **G** Jojosi 1, RG 1 with *n* = 3 artefacts with laminar production sequences in a unidirectional knapping sequence, repeatedly detaching *débordant* flakes.

All assemblages are characterised by abundant small knapping debris and unretouched blanks with some cores and an absence of formal tools like unifacial points (see Supplementary Note 4). The lack of small retouching flakes illustrates that retouching did not take place on-site. Noticeable features include the high proportion of cortical pieces (33–62%) and of elements from core preparation and rejuvenation (Supplementary Tables 30–44). Potential end products, such as large blades or preferential flakes, are rare. Use-wear analyses of 40 non-cortical blanks from all lenses identified only a single piece with potential traces of use (Supplementary Note 5). In terms of technology, all assemblages fit into the MSA, without any microlithic technologies or handaxes (Fig. 5). Core reduction at Jojosi 5, 6, and 7 mostly proceeded via platform modalities and are either uni- or bidirectional. Based on a technological reading of the cores and flaked products, the main goal of knapping was the production of large blades and presumably bladelets, though these end products are mostly missing. Levallois technology is only recognisable at the youngest assemblage, Jojosi 1, with both cores and frequent core-edge flakes, but only rare Levallois flakes (see also ref. 42).

In conjunction with the technological and use-wear analysis, the lithic refitting demonstrates that the primary actions the knappers performed at the excavated locations encompassed initial decortification of large hornfels blocks, followed by the preparation and rejuvenation of cores (Figs. 4 and 5). Based on the completeness of the assemblages and the preservation of finds, the rarity of end products from the reduction sequence at all stratified sites, but also amongst the surface finds, can be considered a true negative. Knappers removed their finished products to other places beyond the donga landscape.

The sum of these characteristics identifies all Jojosi locations as combined raw material procurement sites and knapping workshops for the main purpose of (i) obtaining large blocks of hornfels; (ii) reducing them to produce various unretouched blanks, and (iii) taking the resulting blanks away for future use beyond the Jojosi landscape.

## Discussion

Jojosi provides a uniquely well-preserved archaeological window into patterns of raw material acquisition by early humans. Typically, studies on this key behavioural aspect in the African Stone Age have relied on residential sites in caves and rock shelters with a pronounced palimpsest character and distant from rock sources[20,41,49,50], or had to deal with primary stone outcrops in the landscape that are either devoid of sediments and/or yield only surface finds[28,29]. The former cases present challenges in identifying the origins, transport distance, and specific mode of raw material procurement—this issue also applies to the fewer stratified open-air palimpsests nearby important raw material sources (e.g., refs. 51,52)—whereas the latter lack a secure stratigraphic context, temporal resolution, and an absolute age for such behaviours. At Jojosi, rare, high-resolution insights into Pleistocene material procurement are made possible by an unusual confluence of factors: (i) the existence of large primary and secondary sources of a sought-after raw material that repeatedly attracted past humans to this landscape for >100,000 years; (ii) geomorphic processes that gently buried and cemented human activities by steady low-energy processes; and (iii) the modern erosion of these localities, allowing for their discovery and archaeological access to the relevant sediments. At other sites, comparable artefact-bearing sediments may

**Table 1 | Luminescence age calculations using BayLum and results of high-resolution gamma spectrometry to determine U, Th, and K contents**

| Sample ID | CLL No | $U$ [ppm] | Th [ppm] | $K$ [%] | Total $\dot{D}$ [Gy ka$^{-1}$] | $n_{accepted}$ ($n_{saturated}$) | BayLum Dose [Gy] | BayLum Dose [Gy, 1σ range] | BayLum age [ka] | BayLum age [ka, 1σ range] |
|---|---|---|---|---|---|---|---|---|---|---|
| Jojosi 1 | | | | | | | | | | |
| JOJO-1-1 | C-L5531 | 0.48 ± 0.04 | 2.66 ± 0.19 | 0.30 ± 0.01 | 0.74 ± 0.03 | 30 (0) | 112 | 101–119 | 156 | 141–166 |
| JOJO-1-2 | C-L5532 | 0.42 ± 0.03 | 2.48 ± 0.18 | 0.29 ± 0.01 | 0.74 ± 0.03 | 35 (0) | 96 | 94–99 | 132 | 121–139 |
| JOJO-1-3 | C-L5533 | 0.46 ± 0.03 | 2.65 ± 0.19 | 0.28 ± 0.01 | 0.81 ± 0.03 | 28 (0) | 89.5 | 85.3–91.3 | 111 | 106–117 |
| Jojosi 5 | | | | | | | | | | |
| JOJO-85U | C-L5349 | 0.39 ± 0.03 | 1.97 ± 0.14 | 0.28 ± 0.01 | 0.8 ± 0.03 | 35 (2) | 180 | 163–197 | 218 | 203–242 |
| JOJO-5-4 | C-L5520 | 0.39 ± 0.03 | 2.29 ± 0.17 | 0.29 ± 0.01 | 0.78 ± 0.03 | 34 (0) | 135 | 125–137 | 175 | 160–187 |
| JOJO-5-5 | C-L5521 | 0.41 ± 0.03 | 2.34 ± 0.17 | 0.28 ± 0.01 | 0.81 ± 0.04 | 28 (0) | 119 | 114–125 | 148 | 136–160 |
| Jojosi 6 | | | | | | | | | | |
| JOJO-6-3 | C-L5933 | 0.48 ± 0.04 | 2.45 ± 0.18 | 0.33 ± 0.01 | 0.74 ± 0.05 | 32 (0) | 182 | 170–192 | 242 | 222–258 |
| JOJO-6-5 | C-L5935 | 0.4 ± 0.03 | 1.97 ± 0.14 | 0.27 ± 0.01 | 0.64 ± 0.05 | 30 (0) | 156 | 147–166 | 219 | 201–235 |
| Jojosi 7 | | | | | | | | | | |
| JOJO-7-1 | C-L5936 | 0.58 ± 0.04 | 3.79 ± 0.27 | 0.3 ± 0.01 | 0.91 ± 0.04 | 32 (4) | 205 | 189–221 | 226 | 204–245 |
| Jojosi Triple (TRPL) | | | | | | | | | | |
| JOJO-TRPL-1 | C-L5536 | 0.41 ± 0.03 | 2.85 ± 0.2 | 0.45 ± 0.01 | 0.92 ± 0.04 | 39 (14) | 568 | 546–581 | 622 | 583–654 |
| JOJO-TRPL-2 | C-L5537 | 0.31 ± 0.02 | 1.49 ± 0.11 | 0.24 ± 0.01 | 0.6 ± 0.03 | 35 (1) | 309 | 286–330 | 526 | 479–561 |
| JOJO-TRPL-3 | C-L5538 | 0.43 ± 0.03 | 2.68 ± 0.19 | 0.36 ± 0.01 | 0.82 ± 0.04 | 34 (3) | 212 | 202–224 | 256 | 236–273 |
| JOJO-TRPL-4 | C-L5539 | 0.47 ± 0.03 | 2.82 ± 0.2 | 0.34 ± 0.01 | 0.82 ± 0.03 | 36 (0) | 184 | 177–189 | 236 | 224–246 |
| JOJO-TRPL-5 | C-L5540 | 0.44 ± 0.03 | 2.63 ± 0.19 | 0.29 ± 0.01 | 0.79 ± 0.03 | 30 (2) | 195 | 178–205 | 224 | 213–237 |

Details regarding the internal K concentration, water content, and depth used for external, internal, and cosmic dose rate determination are outlined in Supplementary Note 2. The number of accepted ($n_{accepted}$) and saturated ($n_{saturated}$) aliquots for equivalent dose determination is given in the table; here $n_{accepted}$ includes $n_{saturated}$. For BayLum doses and ages, 1σ ranges are given, which represent the 68% credible interval calculated using BayLum[79]. Whilst the BayLum doses are based on calculations of the individual samples, the ages given were calculated including stratigraphic information.

be buried many metres under the current land surface and are invisible. Jojosi presents an uncommon, fine-grained record of human interactions with a raw material on exceptionally short timescales within a landscape scale that includes among the highest number and proportion of refits of any MSA assemblage in Africa and one of the few open-air sites with comparable resolution in a dated and stratified context in the entire Pleistocene Stone Age record (see refs. 49,53,54).

Human fossils from Florisbad (~260 ka)[31] and Border Cave (~170–80 ka)[30] document the presence of *Homo sapiens* in southern Africa during the late Middle Pleistocene. Based on our combined results from the geomorphological and sedimentary context, luminescence dating, technological, use-wear, and refitting analyses, human groups repeatedly performed short visits to the primary and secondary hornfels sources of the Jojosi dongas over tens of thousands of years from at least ~220 ka until ~110 ka, and potentially even longer. Different from other typical workshop sites of the Middle and Late Pleistocene[28,29,51,55], Jojosi contains no evidence of complete reduction sequences or intense tool production, neither in the stratified sites nor in the surface material. Instead, the main purpose of these visits was to reduce large hornfels blocks and produce blanks from this high-quality raw material before removing them for future use at other locations. Identifying these destinations in the wider landscape of eastern South Africa will be a key endeavour of future fieldwork and of analytical studies, including the development of geochemical tracing of hornfels[28], which could provide deeper insights into the transport distances and destinations of the Jojosi hornfels. Despite being a landscape with diverse resources and a favourable topographic position, we encountered no evidence of long-term occupations or the execution of other activities such as retooling or large-scale hunting of animals at Jojosi, similar to the situation at Taramsa in Egypt[12]. The

Jojosi observations from several stratified contexts reflect the more widespread and abundant surface record of stone tools—and an almost complete lack of fossilised bone—which mostly comprises tested blocks, large cores and large, often cortical, flakes, but few finished products or retouched tools.

Our interpretation of Jojosi as a specialised locality intended for hornfels extraction in the form of usable blanks via strategic expeditions is further strengthened by the fact that other raw materials used widely in the MSA, such as dolerite, quartzite, and quartz, are abundantly available in the immediate landscape (i.e., as river gravels) but have not been used. This targeting of Jojosi hornfels may be explained by its special characteristics, such as the high flaking quality, the extent of the primary outcrop, the presence of frequent large angular blocks that allow for reduction without abundant prior preparation, and the easy production of large blades. Our findings attest to a high degree of planning, long-term anticipation of technological needs, and a knowledgeable survey and selection of sought-after raw materials. These technological abilities of early modern humans during the MSA significantly extend the record from the younger but equally long and only comparable record in Africa of Taramsa in Egypt, where numerous stratified and dated sites show that people targeted and exploited high-quality chert cobbles by digging ditches and pits from ~166 ka until ~60 ka[12].

It is only within the African MSA record that we have obtained sufficient archaeological resolution to securely identify specialised provisioning as the main or single purpose of a site, and can pinpoint the timing and duration of these behaviours. Precursors of direct procurement may well extend into earlier periods. The current ESA record, however, commonly derives from time-averaged surface contexts, remains undated, or does not demonstrate that specific raw

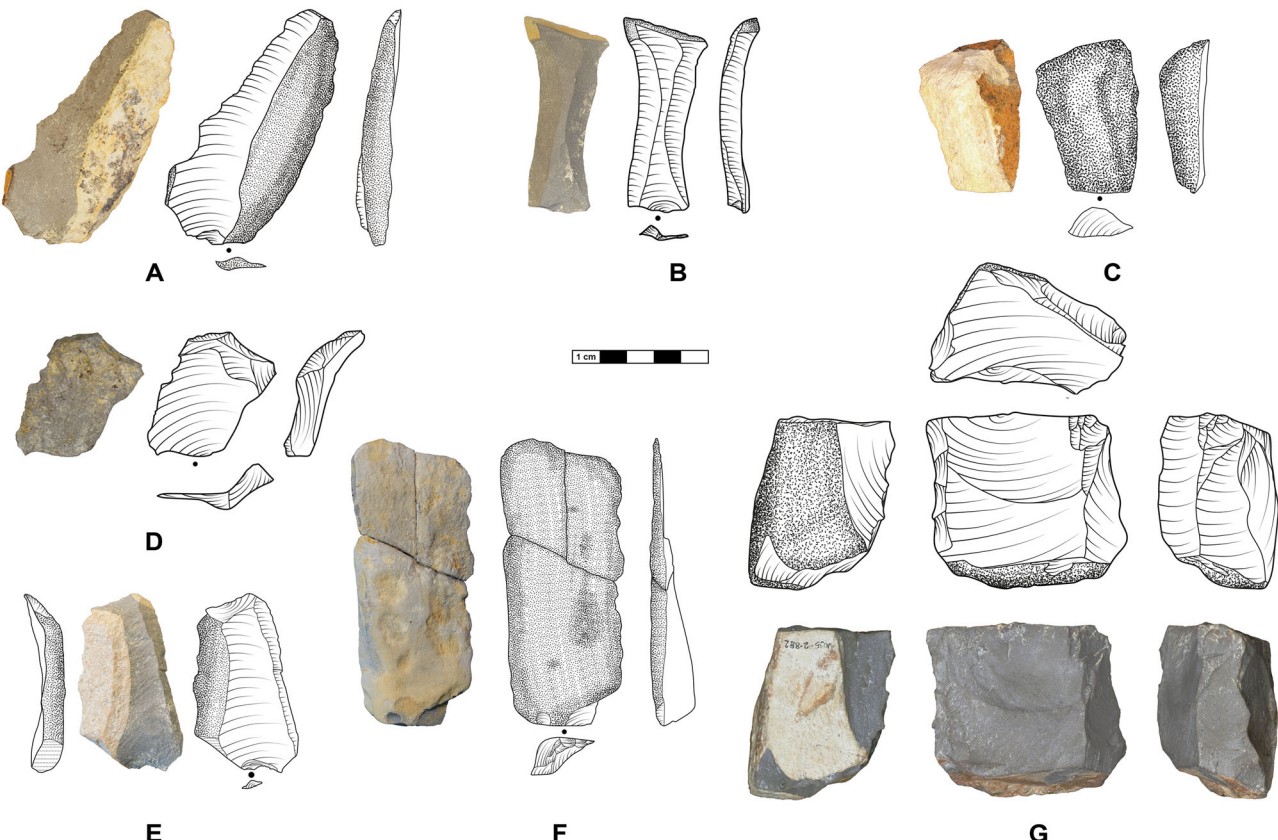

**Fig. 5 | Technologically relevant stone tools from Jojosi 5, 6, and 7.** Depictions in dorsal, lateral, and platform views, scale 1:1. **A** JOJ5-1-802, *débordant* flake with cortical edge. **B** JOJ5-1-771, plunging blade with distal cortical edge. **C** JOJ6-1-127, fully cortical flake. **D** JOJ5-2-905, *débordant à dos limité* flake, scale 2:1 for enhanced visibility. **E** JOJ6-2-573, *débordant* flake with lateral cortical edge. **F** JOJ7-1-92/121/126 as Refit Group 55, almost completely cortical blade. **G** JOJ5-2-882, Multi-platform core.

material collection was planned ahead and separated from other tasks carried out in the wider landscape[28,56,57]. As a regional comparative example, the late Acheulean open-air site of Wonderboom demonstrates the removal of flakes directly from outcropping quartzite bedrock and large blocks in the landscape deriving from bedrock ('flake harvesting') at eight localities that appear broadly similar to Jojosi in their focus on one raw material and the repeated visit to this raw material source. Yet, Wonderboom also features complete reduction sequences and tool production in the form of handaxes, and further connected activities cannot be ruled out due to a lack of stratigraphic resolution and complex site-formation processes[55].

The findings at Jojosi demonstrate that under specific circumstances, groups of *Homo sapiens* repeatedly practised specialised procurement of lithic resources already well into the Middle Pleistocene and over long durations at a single locale. The nature and magnitude of these activities challenge the status quo assumption of predominantly embedded procurement of rock resources in the Pleistocene, which comes from a perspective that emphasises efficiency and optimal foraging[3,4]. Extensive surface extraction sites for raw materials from the MSA, Middle Palaeolithic, and potentially earlier in the Near East, northern and southern Africa, provide further circumstantial evidence for direct provisioning that could extend to other hominin species[55,56,58]. A further bias to prevailing views may lie in the difficulty of distinguishing between embedded and direct procurement in the archaeological record due to the much higher amount of evidence and resolution required for demonstrating the latter behaviour. In contrast to dominant assumptions, some ethnographic work has pointed out the importance of other functional and socio-cultural factors in obtaining tool stones for recent hunter-gatherer groups. Specific quarries and rocks may be associated with a special history, territory, belief, or meaning[59,60], and the inter-generational use of specific sources may start an autocatalytic feedback process in which past traces of raw material extraction create visible incentives for further exploitation[61]. The identification of double patina on some hornfels artefacts in the stratified and surface artefacts at Jojosi, coupled with the sheer quantity of flaked pieces on the surfaces, and the long duration of use, suggests that this landscape became a known and focal point over time for intentional, repeated visits. The result is a landscape transformed by both geological forces and human agency, evidencing incipient niche construction already in the late Middle Pleistocene in the form of re-distribution of (lithic) resources, reconfiguration of landscapes, and re-use of constructed locales. Here, Jojosi joins comparable (undated) stone tool-related activities of early modern humans in Africa[58] and other niche construction behaviours such as the over-exploitation of resources[62] or fire use linked to shifting patterns of vegetation and erosion[63].

Our findings add important insights to ongoing research on *Homo sapiens* in the MSA of Africa and their ability for exceptional behavioural flexibility and long-term planning stretching back to the Middle Pleistocene. The making of shell beads[64], long-distance transport and aggregation of lithic[19] and other materials[65], hafting[66], heat-treatment[67], and ochre use[68] now all predate the Late Pleistocene. The MSA record increasingly demonstrates the deep roots of innovative, diverse material culture use by *Homo sapiens*, close to their biological origin. While by no means showing a simple linear or cumulative pattern[69], the record of Jojosi and the archaeology of human origins more broadly testify to a core capacity of our species that lies in its adaptive flexibility and behavioural plasticity when

engaging with the material world and its affordances that characterise all living people. This capacity encompasses activities as disparate as the ways of how to acquire certain raw materials, as shown here, but also in transforming[67], combining[70], using[71], and imbuing them with meaning[72]. Jojosi adds an important puzzle piece to this story and invites further search into both the provisioning of diverse materials as the basis of Pleistocene human technology, a stronger emphasis on the multiple ways in which humans interacted with their (self-made) material worlds, and a renewed interest in open-air localities that remain heavily under-researched in the MSA of Africa but may provide unique insights under the right circumstances.

## Methods

### Ethics and Inclusion

For the archaeological excavations and collection of archaeological material, the relevant permits were issued by the local heritage agency AMAFA (PermitID: 3848 REF: SAH22/18276; 3850 REF: SAH22/18276) to M. Will, valid from 05/12/2022 to 05/12/2025. The permit for analysis and temporary export of the Jojosi 5 & 6 stone tools was issued by AMAFA (PermitID: 3989 REF: SAH23/21517 & 23/087) and SAHRA (CaseID: 22070; PermitID: 3968) to M. Will. All exported artefacts were returned to the KwaZulu-Natal Museum in Pietermaritzburg in January 2025. As excavation took place in a traditional authority area, we first received permission to conduct our work from Morena Molefe of Batlokoa Ba Molefe and the respective Tribal Council. During each fieldwork season, we presented our past and ongoing work in meetings to the Tribal Council. After the completion of the fieldwork in the Jojosi dongas in 2025, printed bilingual posters (in English and Zulu) were passed on to the Tribal Council that communicate the central findings of this project as well as additional educational resources on archaeology. These posters will be distributed to the local schools and community centre.

### Overview

Our multidisciplinary study combines geography and geology (Supplementary Note 1), luminescence dating (Supplementary Note 2), archaeology and excavation (Supplementary Note 3), lithic analysis (Supplementary Note 4), use-wear studies (Supplementary Note 5), and zooarchaeology (Supplementary Note 6). The methods used are described below, with further details and contextual information in the Supplementary Information.

### Excavation

We adapted archaeological excavations to the complex sediment geometry and the specific kinds of archaeological occurrences in the dongas, neither allowing large-scale digging of horizontal planes as is usually done in an open-air setting. Instead, excavations consist of multiple, targeted explorations of outcropping archaeological material in small areas within the often non-contiguous sediment bodies. These occurrences receive individual numbers in ascending order (e.g., Jojosi 5) and are analytically treated as separate sites and assemblages. Within these sites, we encountered individual features of lithic accumulations which we termed lenses. Jojosi 5 consists of two lenses, Jojosi 6 consists of two lenses, and Jojosi 7 consists of one lens. We measured the encountered artefacts >2 cm in size in 3D with a total station and an EDM programme in a local grid system associated with an Access database (e.g., ref. 73). We recorded the orientation of elongated artefacts >2 cm via Total Station by two measurements at the opposite ends of the piece. All sediments were screened through a sieve of 10 mm and 1 mm to recover smaller archaeological finds, except for Jojosi 7, where wet sediments allowed only for non-systematic hand collection of small material in the field (see Supplementary Note 2). The material is stored and curated in the KZN Museum in Pietermaritzburg. Supplementary Note 2 provides additional excavation details and field photographs of the ongoing

excavations and artefact lenses for Jojosi 5, 6, and 7, as well as photographic material for Jojosi 1 from the 1991 excavation (Supplementary Note 3).

### Geography and geology

The composite stratigraphy of the succession of accretionary hillslope deposits was mapped from gully sidewall exposures to define the relationships between infilled palaeogullies and colluvial deposits. We documented six sedimentary profiles: two representing the geoscientific reference profile of the site (Jojosi Triple Junction) and four describing the sedimentary succession at archaeological sites (Jojosi 1, Jojosi 5, Jojosi 6, and Jojosi 7). Stratigraphic classification followed the system of Botha[33], which has proven effective for establishing a framework for Late Pleistocene sedimentary deposits of the Masotcheni Formation in KwaZulu-Natal. This approach integrates allostratigraphic and pedostratigraphic techniques to capture both erosional and depositional histories of rock units and the development of pedoderms.

During the 2022–2024 field seasons, we acquired multiple UAV datasets using DJI Phantom 4 Pro and DJI Air 2 platforms to produce annual high-resolution photogrammetric products. We used the Structure-from-Motion software Agisoft Metashape 2.2.1 to create Orthophotos with a spatial resolution of up to 2 cm, Digital Surface Models with a spatial resolution of up to 10 cm, and 3D Models. The resulting data were used to create spatial maps for surveying archaeological sites and stratigraphic contacts. Furthermore, we produced orthographic profile sections of gully sidewalls to assist in the mapping and correlation of stratigraphic units over extended distances. Geospatial coordinates of archaeological sites, geoscientific features, and ground control points for stereophotogrammetry were recorded using a Real-Time Kinematic (RTK) differential GPS (ZED-F9P by ArduSimple), receiving correction signals from the Newcastle, Greytown, and Ulundi reference stations via the South African GNSS base station network TrigNet.

Six detailed profile descriptions with 23 sediment texture and chemistry samples (Supplementary Tables 1 and 2), as well as 20 mineralogical samples analysed with semi-quantitative X-ray diffraction (XRD) analysis (Supplementary Table 3), contextualise the archaeological deposits. Texture and chemical analyses (full-fraction) were performed at the Soil Science Laboratory, CEDARA College of Agriculture, Hilton, South Africa. Mineralogical composition was determined using semi-quantitative XRD at the XRD Laboratory, Council for Geoscience, Pretoria, South Africa.

### Luminescence dating

Fourteen luminescence samples were collected in total by hammering opaque stainless-steel tubes into cleaned outcrop surfaces or by carving blocks from exposed outcrops. For dose rate determination, additional samples were taken from the sediment surrounding the luminescence samples. Sample preparation for equivalent dose and for dose rate determination was conducted in the Cologne Luminescence Laboratory (CLL) at the University of Cologne. Luminescence measurements were carried out at the CLL and at Risø (Technical University of Denmark, DTU).

High-resolution gamma spectrometry and beta counting were used to determine the sediment dose rate delivered to the samples, and for internal dose rate determination, respectively. Variability in U, Th, and K concentrations in layers influencing some of the luminescence samples necessitated scaling the gamma dose rate following Aitken[74]. The Dose Rate and Age Calculator (DRAC[75]) was used to calculate environmental dose rates for each luminescence sample.

For luminescence dating, sand-sized feldspar grains (200–250 µm) were retrieved from the samples under red light conditions using chemical treatments, sieving, and heavy liquid density separation. Luminescence measurements of small multi-grain aliquots

(1 mm diameter) were performed using Risø luminescence instruments (e.g., ref. [76]). Luminescence measurements were performed following a post-infrared infra-red stimulated luminescence (post-IR IRSL$_{225}$) protocol[77,78]. The protocol was defined and tested in a separate study, which also evaluated the use of feldspar multi-grain aliquots and single grains at Jojosi[47]. Further details are given in the supplementary material.

The low dose rate environment (~0.8–0.9 Gy ka$^{-1}$) of Jojosi enabled dating of sediments back to ~600 ka. Luminescence ages were calculated using BayLum[79]. This Bayesian hierarchical approach allowed us to include (i) information provided by saturated grains, and (ii) stratigraphic information in our age calculations.

Luminescence results of samples retrieved from outcrops to constrain the general palaeoenvironmental setting indicate that cut and fill processes shaped today's donga landscape since at least 583–654 ka (JOJO-TRPL-1), with multiple phases of cut and fill commencing until at least 106–117 ka (JOJO-1-3; see Fig. 2).

Luminescence samples taken at Jojosi 1, Jojosi 5, Jojosi 6, and Jojosi 7 bracket the timing of the archaeological occurrences and human tool stone procurement. Samples collected at each of these four sites are in stratigraphic order for each site individually. The artefact lenses embedded in the profiles can be constrained to 106–139 ka at Jojosi 1, 136–187 ka at Jojosi 5, 201–258 ka at Jojosi 6, and shortly after 204–245 ka at Jojosi 7, thus attesting the repeated and persistent procurement of hornfels at Jojosi (see Fig. 2).

## Lithic analysis

A total of 2675 lithic artefacts >2 cm and 27,150 lithic artefacts <2 cm were recovered during the excavations of the stratified sites of Jojosi 1, 5, 6, and 7. For all artefacts >2 cm, we employed standard attribute analyses for a techno-typological assessment, recording various categorical and metrical attributes (see ref. [42]: SI 2) and following standard approaches in the field[80–83]. In addition, a *chaîne opératoire* approach was applied to the entire assemblages to reconstruct core reduction and to infer the position of artefacts (e.g., end products)[84,85]. Refitting proceeded following general guidelines and terminology as set out by Cziesla[86] and as practically described by Sumner[87] and Vaquero et al.[88]. Artefacts were first sorted into smaller raw material units by macroscopic observations such as matches of cortical surface varieties. Targeted refitting was then conducted with each artefact lens laid out in its entirety, considering morphology, size, surface texture, knapping accidents, and reduction stage for the artefacts. The discovered conjoins and refits were then numbered, recorded, temporarily joined together with removable adhesive compound, and photographically documented. All artefacts <2 cm were counted, classified into their size categories (1–5 mm, 6–10 mm; 11–20 mm), and identified by raw material, which in all cases was the local hornfels. More detailed results on the lithic assemblages are provided in Supplementary Note 4.

## Use-wear

We conducted the traceological analysis of the archaeological and experimental material at the Material Culture Laboratory (MCL) at the University of Tübingen. We performed the traceological analysis combining low- and high-power approaches, following a well-established methodology used in functional studies[89–94]. We started by scrutinising the lithic artefacts under an Olympus SZX7 stereomicroscope, equipped with a magnification range of 8× to 56× and an LED ring light source, as well as external optical fibres. This initial observation allowed an assessment of the preservation state of the lithic surfaces, as well as the characteristics and distribution patterns of edge damage and rounding. When use-related edge damage and rounding were identified, we evaluated the use motions and the hardness of the worked materials. Items showing use-related edge damage were further examined at higher magnifications using a BX53M metallographic microscope equipped with vertical incident illumination, which enabled observations of up to 500× to detect microwear features, including polish, striations, and micro-rounding. The artefacts were consistently handled throughout the analytical process with powder-free nitrile gloves. When necessary, specimens with sediment particles were gently cleaned in a bath of demineralised water and, when required, soaked in an ultrasonic tank.

We based the use-wear interpretations of 40 selected archaeological specimens (see Supplementary Table 45) on the results of a systematic experimental reference collection specifically designed for this case study. Initial observations of the archaeological sample showed a very low frequency of flakes or blades that might have been used, as most tools displayed sharp, unmodified edges. This result led us to focus the reference collection on a selection of materials and activities that could have been performed within the context of the dongas. We knapped a total of 27 flakes and blades from hornfels blocks collected near the sites. We documented the active edges of the replicas in profile, shape, and cross-section, and measured the edge angle before use. We also took pictures of the active edge before use and compared the same area after use to observe the development of macro and micro traces on hornfels (Supplementary Fig. 49). We tested activities compatible with the straight morphology of the flakes and blades recovered from the sites. We carried out longitudinal unidirectional (cutting) and bidirectional (sawing) activities, as well as transverse unidirectional (whittling, debarking) and bidirectional (scraping) activities. The plant material consisted of fresh and dry wood, while the processed animal material included meat, skin, and bone. All the replicas were used only to process one material for a total of 60 minutes (Supplementary Fig. 50). All experiments were conducted under controlled conditions, and attributes were documented on an experiment sheet for each replica. The main experimental variables are listed in Supplementary Table 46.

Additional comparisons were conducted using the reference collection available at the MCL, including chert and flint flakes and tools used to process the same materials as the hornfels replica, and for the same duration. This enabled us to observe trends in macro- and micro-trace formation on hornfels compared with more resistant raw materials. Hornfels has not been extensively studied regarding wear mechanisms, and comparisons with other studies are limited (but see ref. [46]). However, our preliminary observations indicate that hornfels is more susceptible to abrasion than cryptocrystalline rocks. This impacts the appearance and development of polish, microscarring, striations, and rounding, leading to rapid surface changes.

## Zooarchaeology

The faunal material retrieved from Jojosi 7, Lens 1, is too fragmented to be identified taxonomically beyond the family level. We used the bovid size class system proposed by Brain[95], as well as the modern distribution and habitat preferences of southern African bovids presented in Skinner and Chimimba[96], to suggest a list of bovid taxa possibly occurring in the area. All faunal remains were investigated systematically under a binocular microscope, and both biotic and abiotic surface modifications were recorded following criteria defined in the literature (e.g., refs. [97,98]). Weathering stage attribution follows Behrensmeyer[99]; burning intensity uses the colour-code proposed by Stiner et al.[100].

## Reporting summary

Further information on research design is available in the Nature Portfolio Reporting Summary linked to this article.

# Data availability

The data generated in this study are provided in the main article and the Supplementary Information. Source data are provided as a Source Data file. All archaeological material of this study is permanently stored

at the KwaZulu-Natal Museum in Pietermaritzburg, 237 Jabu Ndlovu St., South Africa, with access via the Principal Curator, Dr. Geoffrey Blundell (gblundell@nmsa.org.za). Source data are provided with this paper.

## Code availability
No code was used for this study.

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

## Acknowledgements

This work would not have been possible without the unwavering support of Morena Molefe of Batlokoa Ba Molefe and the Tribal Council, who permitted us to conduct research in their traditional authority area and provided ongoing support for our work. We are indebted to Geoffrey Blundell, Gavin Whitelaw, and the staff of the KwaZulu-Natal Museum in Pietermaritzburg for their constant help in accessing collections and literature, and for providing working space. Our gratitude goes to the KwaZulu-Natal Amafa and Research Institute and the South African Heritage Resources Agency for providing the permits for our excavation and analyses. We would like to thank Prof. Nicholas J. Conard and Prof. Volker Hochschild for their support with laboratory and field equipment and facilities. We are also grateful to Dr. Anja M. Zander for performing high-resolution gamma spectrometry measurements. We would like to thank the student assistants, Leah Böttger, Anna Brückner, Andreas Peffeköver, Hanna Pehnert, Nothando Shabalala, Valentin Sorg and Felix Weinschenk for their commitment in the field and the laboratory. MW was funded by the Deutsche Forschungsgemeinschaft (DFG, German Research Foundation) - Project number 467042592. SR received funding from the European Union's Horizon Europe research and innovation programme under the Marie Skłodowska-Curie grant agreement No 101103587. Further funding was received by MW and CS from the Heidelberg Academy of Sciences and Humanities in the context of the research project "The Role of Culture in Early Expansions of Humans". We acknowledge support by the Open Access Publishing Fund of the University of Tübingen.

## Author contributions

Conceptualisation: M.W. Project methodology: M.W., C.S., S.R., G.H.D.M., and G.A.B. Investigation: all authors contributed in the field or in the laboratory. Funding acquisition: M.W. Project administration: M.W. and G.H.D.M. Supervision: M.W. Writing (original draft): M.W., C.S., S.R., and G.H.D.M. Writing (review and editing): M.W., C.S., G.D.H.M., G.A.B., M.A.B., L.M., A.M., A.V., F.V., S.R.

## Funding

## Competing interests

The authors declare no competing interests.
