## [Transparent Peer Review file · Nature Communications]

Specialised and persistent raw material procurement by humans in the Middle Pleistocene

Corresponding Author: Dr Manuel Will

Version 0:

Reviewer comments:

Reviewer #1

(Remarks to the Author)

I have read with interest the manuscript of Will et al relating to the discovery and excavation of multiple localities at Jojosi in Kwazulu-Natal. I have to say I enjoyed reading the paper and found it overall to be an accomplished exposition of the kind of field and laboratory analysis that pushes forward the field of Stone Age archaeology. The sites at Jojosi certainly seem to represent a different behavioural signature than is typically found at Mid-Pleistocene sites in Africa, and the presence of discrete accumulations of well preserved and refitting lithics, I agree, warrants close study and publication. The results, as presented, are overall reasonable, and I consider that the authors have done a sound practical and technical job at in recovering the data and also in lab analysis and interpretation (although please note below comments regarding some descriptive and potentially worrying issues with the lithic use-wear analysis). The data and fieldwork is well presented and clear (although please see below for a non-exhaustive list of minor additions to the Figures and other information provided in the Supplementary Information). I will be interested to know more about the sites when the researchers are ready to conduct further excavations across the wider landscape, as they suggest they are planning.

I believe that this paper should be published, but I have some comments and concerns that I would like to see addressed before it is accepted for publication. These cover some significant aspects but also some more minor issues which could be considered altogether 'minor corrections'. Beforehand, it is important to clarify that my expertise is in African Stone Age lithic analysis and field excavations, but not in OSL dating. It is extremely important then, to ensure that one of the reviewers of this paper has dedicated expertise in that domain.

Main concerns:

My main issue with this manuscript is not in the technical results presented or the data interpretation, but with the arguments presented for the importance of these sites and their cognitive uniqueness/importance. Breaking this down further, I agree that the sites represent a rare occurrence of in situ flaking of a single material that also seems to exclusively represent the early stages of lithic production. In that sense, these sites are special, and warrant close attention. However, I am not particularly persuaded by the idea that a) Jojosi sites certainly reflect a targeted (direct) provisioning behaviour, b) that such a behaviour is considered by stone age archaeologists working in the Mid-Pleistocene to be 'non-standard' or 'not to have been undertaken', or even 'not proven', and furthermore, c) that such a behaviour reflects the cognitive significance attributed to it by the authors. Starting with point a), the authors themselves suggest that the sites do not 'prove' a direct provisioning behaviour. Beyond the semantics and questioning over what could constitute sufficient proof, the point made by the authors that the preservation of Jojosi and time-averaged nature of most other mid-Pleistocene sites shows they are aware that the truly special thing about these sites is their snapshot of brief technical behaviours, and that other African sites may have accumulated as a result of direct provisioning, but their time-averaged nature means we cannot see it. If my reading of the manuscript and authors point here is correct, then I have to say I agree with them. While Jojosi does constitute a form of 'uniqueness', it is also not very exciting as a finding if we agree that other sites reflect the same behaviour but that this behaviour is masked by their lack of chronological control. Turning to b), I myself have never considered that direct provisioning would not have been undertaken by mid Pleistocene hominins, which is partly perhaps why I feel the discovery of such a behaviour is not the most exciting thing about Jojosi. In fact, I would go further, and suggest that 'proving' direct provisioning of lithic resources and considering it to be significant indicates a kind of unwarranted behavioural conservatism on the part of the authors. To me, this feels like a discovery that the authors made and then struggled to search for its

cognitive significance. With point c), I consider based on wider sites and datasets (some of which the authors do discuss) about transport and selection behaviours for this time range, that (even if we agree for a second that Jojosi does indeed 'prove' such a behaviour) demonstrating direct provisioning does not reflect the discovery of a new cognitive threshold on behalf of mid-Pleistocene hominins. As an aside, missed in this paper is Barham's publications on the movement of pigments at Twin Rivers, Zambia, which demonstrate a long-term commitment to the aggregation of minerals at >200 Ka (similar to Jojosi).

I think overall the authors need to reconsider whether the identification of direct provisioning is truly surprising, especially when the flaking was done quite close to the hornfels source – I am not at all surprised that mid-Pleistocene hominins were capable and routinely went to quarries and removed materials to nearby safe places for knapping. I would be surprised indeed if this were NOT the case.

Reading the paper, I was led to think of other examples of sites with similarities to Jojosi. While I admit I cannot myself think of site in such a good state of preservation, I think the authors need to include and consider the relative importance of the sites of Gademotta and Kulkuletti (279 ka) in Ethiopia, which are older than Jojosi, and show the specific and repeated exploitation of Obsidian from a source that is further away than the Jojosi source is from the sites there. While at Gademotta other materials are represented in the assemblages, over 94% of all lithics (including debris) are local obsidian, according to Sahle (2013). Of course, Gademotta includes finished artefacts and lacks the spatial and chronological preservation of Jojosi, but I would find it remarkable that the authors do not consider Gademotta as an example of special lithic provisioning, and also likely/probably directly acquisition, as they argue for Jojosi.

Gademotta then, at least needs inclusion as a reference point and comparison regarding its potential direct provisioning and cognitive importance, potentially to illustrate the special importance of Jojosi. Furthermore, I think the way that the paper is written, it takes some work on the part of the reader to tease out exactly what the unique situation is at Jojosi. It is not directly stated, and so it takes some time to figure out that it is not the distance from source, and it is not that Jojosi itself is an actual quarry....these were my first assumptions and were never directly countered. The authors could better clarify upfront that Jojosi is a secondary location used for the decortication/early stages of lithic production, and seems to have been exclusively used for this purpose. All this draws attention to the need for greater clarity over the theoretical arguments being made here by the authors. Refining these arguments will help demonstrate the special contribution of Jojosi to cognitive debates. Overall, as stated, I think this paper is important, but I do think it would benefit from a more critical eye on the part of the authors concerning their theoretical contextualisation of the sites, and their consideration of direct provisioning as an exciting and cognitively significant finding in the mid-Pleistocene.

Finally, I think that some more images of the lithics in the main paper (individual pieces and not only refits as at present) would be interesting and are warranted in light of the importance being claimed for the site. Let us see some technological features rather than just the refits, please.

Additional points regarding the lithic use-wear analysis.

The description of 'Ad-hoc experiments' for the use-wear does not sound structured and brings uncertainty to the results described. I am left wondering if the work was done in a serious manner. It is lucky then that the positive use-wear identifications were so few, as otherwise I think this expression would be grounds to throw out any results presented. Please amend the description and ensure future studies are correctly and adequately structured.

Line 632 "additional comparisons were made with non-hornfels materials". Greater specificity is needed on the number and type of comparisons drawn – what were the uses of the wider experimental collection used as a reference point?

Minor but important issues to be addressed:

Line 42: the authors give a date range of 3.0 – 2.6 Myra for the earliest known artefacts. Lomekwi 3 (Harmand et al 2015, as referenced by the authors to substantiate the lower end of this date range), is well dated to 3.3 Ma. 300,000 years is a significant chronological error: if applied to Jojosi we could equally say the site is 0.0 Ma! The stated date range should therefore be amended to 3.3 – 2.6 Myra.

Please consider amending use of the word *débitage*, when what is meant is debris, or more specifically knapping shatter and fragments. Ditto use of the word 'micro débitage'. *Débitage* used as a noun to refer to such material is an inaccurate Anglicisation, even if the misuse of the word is now widespread.

SI 3, Text S2: the meaning of the word 'transects' is not very clear. I understand the rest of the text, regarding the differentiation of contexts and spits etc, but are these transects are spatial delineation? If possible an illustration of what is meant could be useful here, to show the system of excavation. If not, then for sure greater clarity of explanation is required here as the use of several words with potentially overlapping or indistinct meaning creates some confusion. Addendum: the transects are shown in a photo later in the supplementary information, and this photo or similar could be moved or used to illustrate what is meant by transect on first use.

Figures S1-S6: For all the widest-view images, it is important to clarify the scale, either within the images next to the measure seen in the image, or in the figure captions.

Figures S9 and S10. I don't find the combination of x-axis for depth and y for chronology clear at all. It would be much better to show both depth and chronology on the same axis, which would allow readers to grasp the date range for each sample, and how these relate to the position of the samples and the depth range of artefacts at each locality. At the present time, it looks like artefacts might be spread throughout MIS5-8 at each locality, which is erroneous and confusing. Figure S10 is similar and needs to be checked for logic and clarity – although noted in the caption, the use of grey for shading makes an implied confusion with Figure S9 and indicates, contrary to the descriptions of the sequences, that artefacts are present at the Triple Junction.

Figure S19: I do not see the hammerstone being referred to in the caption anywhere, even in the top left as described.

Please highlight or reconsider description. Addendum: I later see the hammerstone marked in Figure S21, but it does not immediately appear as a hammerstone from the 2D visual, so I suggest in both images this artefact is highlighted somehow if the authors want to draw attention to the piece. In S19 it is not located on the top left as described.

Figure S20: same concentration but a different orientation and the hammerstone is still said to be in the top left of the image

but no hammerstone is clearly identifiable. Addendum: I later see the hammerstone marked in Figure S21, but it does not immediately appear as a hammerstone from the 2D visual, so I suggest in both images this artefact is highlighted somehow if the authors want to draw attention to the piece

Reviewer #2

(Remarks to the Author)

Please see my attached review. I consider my comments to be 'minor revisions'.

Reviewer #3

(Remarks to the Author)

Thank you for the opportunity to review this manuscript. The study provides a description of excavations at Jojosi, an open-air MSA archaeological site in KwaZulu-Natal, and considers the site's implications for Pleistocene lithic procurement strategies. In particular, the authors argue that dense accumulations of hornfels artifacts found on eroding surfaces as well as in situ are evidence of a direct procurement strategy, whereby a lithic source is specifically targeted for exploitation. This stands in contrast to embedded strategies that include procurement as part of a wider pattern of movement and resource use; these are more well-known in ethnographic accounts and presumed common in the generation of archaeological patterning as well. The study finds that in situ artifacts show evidence of accumulation with little evidence of post-depositional disturbance; lithics accumulated during the MSA and sites were used on multiple occasions; lithic use is exclusively restricted to the local hornfels and reflects a negative pattern consistent with early stages of core reduction and blank preparation but missing larger flake blanks; and evidence for activities other than lithic reduction is limited. This points to a pattern of regular exploitation of local hornfels and transport of desirable products away from the site, suggestive of direct procurement in the deep past.

Overall, this paper was a pleasure to read. It is clearly written and the study is well-conceived. The findings related to the use of raw material challenge conventional understandings of MSA resource use and will be of significant value to those studying human paleoecology and technological organization. The authors marshal an impressive body of evidence from geomorphology, site formation, archaeochronometry, lithic analysis and refitting, and zooarchaeology. I have no major criticisms of the manuscript or the methods, the suggestions below are offered to clarify some minor items:

-The landscape is described as resource rich (ln 129 and ln 316), but this seems to be an assessment primarily of conditions in the present. The passages and supplement reference a stream as well as the lithic source, but while the latter was evidently accessible in the past it's not clear to me what condition the stream was in, or any other aspect of the environment for that matter, during the timeframes of interest. I find a mixed bag when I look at paleoclimate reconstructions from the region, with Toffolo et al (2017) suggesting a dry phase around 240-200k at nearby Florisbad. I suggest adding a reference or two to better justify the claim of resource richness in the past.

-The lithic accumulations at Jojosi are characterized as being generated over "tens of thousands of years" (ln 31, 92, 233), suggesting a long term accumulation and/or frequent visitation. Based on the dating (ln 560-591), it seems clear that the site was visited more than once over such a window, but the number of separate visits is not fully clear given overlaps in the date ranges. The Jojosi-1 dates, at 1 sd, provide the best evidence for an extended period of accumulation, but actual duration over which any of these assemblages formed within their accumulation windows is necessarily ambiguous. While I believe the authors are probably correct in their interpretation, I suggest rephrasing some of these passages to account for alternative interpretations of the chronology.

-Table S8 is missing its summed totals. Also, I'll admit the densities of 2,000,000 artifacts per cubic meter raised my eyebrow. Extrapolating values from 0.003 m³ excavated to a 1 m³ comparative standard seems like a less-than-intuitive measure, though I'm not exactly sure how to handle this for comparisons sake. I understand that a smaller reference volume is not conventional, but perhaps something like dm³ would be more appropriate. Alternatively, since the table caption already mentions relative orders of magnitude between overburden and artifact lenses, I wonder if it might be more useful here to express these densities as multiples relative to the respective overburdens (e.g, Jojosi-6 Lens 2 = 1433 x overburden)?

Beyond this, I have no substantial criticisms or comments for this manuscript, and believe it should proceed to publication.

References

Toffolo, Michael B., James S. Brink, Cornie van Huyssteen, and Francesco Berna. "A microstratigraphic reevaluation of the Florisbad spring site, Free State Province, South Africa: Formation processes and paleoenvironment." *Geoarchaeology* 32, no. 4 (2017): 456-478.

Reviewer #4

(Remarks to the Author)

Will et al present an interesting study on Jojosi, a raw material workshop dated to the Mid-Pleistocene in the Middle Stone (~220 ka to ~110 ka). The authors show that it represents an important source of hornfels in the region, arguing that it 'contradicts the standard model of embedded procurement for this period' by suggesting long-distance stone procurement of

specific stone (hornfels). The project's team has conducted extensive surveys of the region and done detailed analyses of the lithics and geoarchaeology at the Jojosi sites. The type of lithic workshop discovered at Jojosi is rare and Will et al have done an admirable task of assessing the data and highlighting the significance of the site. I found the paper to be well-written and convincing with good use of available data. I have no major issues with the manuscript but I have one comment. It would be interesting to see how the Jojosi sites relate to other lithic workshops and flake-harvesting localities in southern Africa. I'm thinking particularly of Wonderboom near Pretoria in South Africa. Although that is an Acheulean site, I think it could be of interest to this project in understanding what Will et al call 'tool stone' provisioning. Like Jojosi, it too had little evidence of fauna suggesting it was a specific lithic workshop, and, similar to Jojosi, the focus of the workshop was specifically on one type of material – in that case, quartzite. Indeed, the fact that flake and lithic resource extraction sites and workshops occurred in this region in the Mid Pleistocene (preceding modern Homo sapiens) would add an interesting dimension to what is found at the Jojosi sites.

Minor edits:

Ln 45. Change 'central' to 'key'.

Ln 54-59: Include a few references for these modes.

Ln 338. Replace 'well into' with 'from'.

Version 1:

Reviewer comments:

Reviewer #1

(Remarks to the Author)

I have read the revised manuscripts provided by Will et al. and the replies to the reviewers. The authors have done a detailed and balanced job in addressing my concerns and those of the other reviewers. I am particularly pleased that they were able to clarify better the precise behavioural signature they have identified at Jojosi and their argument for its wider evolutionary significance, as well as changes to some of the figures in the main paper and to the description of the use-wear analysis, which now appears as a well-developed and professionally conducted part of the overall study.

I think that the results of this study are noteworthy, and the claims are supported by the data presented. I recommend this manuscript for publication and pass my congratulations to the research team.

REPLY TO REVIEWERS JOJOSI BIG PAPER

Reviewer #1 (Remarks to the Author):

Main concerns:

My main issue with this manuscript is not in the technical results presented or the data interpretation, but with the arguments presented for the importance of these sites and their cognitive uniqueness/importance. Breaking this down further, I agree that the sites represent a rare occurrence of in situ flaking of a single material that also seems to exclusively represent the early stages of lithic production. In that sense, these sites are special, and warrant close attention. However, I am not particularly persuaded by the idea that a) Jojosi sites certainly reflect a targeted (direct) provisioning behaviour, b) that such a behaviour is considered by stone age archaeologists working in the Mid-Pleistocene to be 'non-standard' or 'not to have been undertaken', or even 'not proven', and furthermore, c) that such a behaviour reflects the cognitive significance attributed to it by the authors.

We thank the reviewer for the general praise of our article, the detailed review of our manuscript and supplement, and the constructive suggestions on how to further improve the paper. Below, we reply to the three major points of criticism a-c) in a more detailed manner.

Starting with point a), the authors themselves suggest that the sites do not 'prove' a direct provisioning behaviour. Beyond the semantics and questioning over what could constitute sufficient proof, the point made by the authors that the preservation of Jojosi and time-averaged nature of most other mid-Pleistocene sites shows they are aware that the truly special thing about these sites is their snapshot of brief technical behaviours, and that other African sites may have accumulated as a result of direct provisioning, but their time-averaged nature means we cannot see it. If my reading of the manuscript and authors point here is correct, then I have to say I agree with them. While Jojosi does constitute a form of 'uniqueness', it is also not very exciting as a finding if we agree that other sites reflect the same behaviour, but that this behaviour is masked by their lack of chronological control.

Our main argument is that at Jojosi, we have the best-demonstrated early and enduring case of direct procurement in the African Stone Age and for *Homo sapiens*. And indeed, these sites preserve much more highly resolved snapshots compared to virtually all other comparable sites. We would argue, like the reviewer, that there are other cases evidencing this behavior, but they are rare and not fully comparable (e.g., Taramsa), and/or heavily time-averaged and/or not stratified and dated (e.g., flint extraction from the Middle Palaeolithic of the Near East or Acheulean workshop sites such as the newly added site of Wonderboom from South Africa, see below). Indeed, many Mid-Pleistocene workshops show a full reduction sequence and intense production of tools, which we do not see at Jojosi, but this does not appear to be purely a result of time-averaging or different resolution, as the pattern is the same in the stratified and surface finds (modified Lines 337-340; 348-351). Ultimately, Jojosi provides a new, secure, and high-resolution anchoring point to discuss the age, relevance, and frequency of direct provisioning before the Late Pleistocene in humans and other hominins, and we would argue that herein lies the particular significance of our findings. We are not claiming that this is the oldest or a singular case of this behavior, and we now also refer more clearly to some potentially earlier precursors, though commonly undated or unstratified sites. We carefully went through our manuscript text again to make sure the text reflects this main argument (e.g., we modified lines 312-320; and lines 383-388; and added lines 337-342 and lines 365-377).

Turning to b), I myself have never considered that direct provisioning would not have been undertaken by mid Pleistocene hominins, which is partly perhaps why I feel the discovery of such a behaviour is not the most exciting thing about Jojosi. In fact, I would go further, and suggest that 'proving' direct provisioning of lithic resources and considering it to be

significant indicates a kind of unwarranted behavioural conservatism on the part of the authors. To me, this feels like a discovery that the authors made and then struggled to search for its cognitive significance.

We respect the opinion of the reviewer, though we argue that based on the current literature (with key references cited in the Introduction of our main text, though there would be many more), the majority view is a different one, and indeed sees direct procurement as either rare or not as relevant as embedded procurement (or just very difficult to demonstrate and therefore underappreciated and largely ignored). See also similar opinions raised by reviewers #3 and #4 on this point. Ultimately, we don't argue from our own position but from the currently prevailing paradigm in the field, which may be conservative in the sense that the reviewer argues. Most important for us is how to best demonstrate direct procurement (and not just assume or deny it), and the rare nature and resolution of the Jojosi sites provide us with one of these uncommon windows. Combining both stratified and unstratified evidence, we agree with the reviewer and consider it likely that direct procurement was much more prevalent than what the field accepts, but the crux is in proving this and pinpointing it better in time and relevance, which is what our paper's main argument is about. As a response to the reviewer, we adjusted our text to more clearly convey the nuances in our argument in the Introduction and Discussion, as we have spelled it out in this response (Lines 67-71 and 91-94 and lines 316-329; 337-342, 365-377 and 383-388).

In addition, while the cognitive side of things is not our main point, we have been quite careful in our interpretations so far and retain that they indicate some relevant capacities - such as increased long-term planning, which is necessary for this behavior - but reduced and qualified them further in the text (e.g., Discussion Lines 404-408). We are more interested in how this shows an additional dimension to human-material interactions, and we now put this angle in the main focus in the Discussion - the adaptive flexibility and behavioural plasticity when engaging with the material world.

We also added another sentence in our Discussion that we think also reflects some of the points the reviewer is making and that may obscure the current picture on past direct procurement: Lines 386-388: "A further bias to prevailing views may lie in the difficulty of distinguishing between embedded and direct procurement in the archaeological record due to the much higher amount of evidence and resolution required for demonstrating the latter behaviour"

With point c), I consider based on wider sites and datasets (some of which the authors do discuss) about transport and selection behaviours for this time range, that (even if we agree for a second that Jojosi does indeed 'prove' such a behaviour) demonstrating direct provisioning does not reflect the discovery of a new cognitive threshold on behalf of mid-Pleistocene hominins. As an aside, missed in this paper is Barham's publications on the movement of pigments at Twin Rivers, Zambia, which demonstrate a long-term commitment to the aggregation of minerals at >200 Ka (similar to Jojosi).

I think overall the authors need to reconsider whether the identification of direct provisioning is truly surprising, especially when the flaking was done quite close to the hornfels source - I am not at all surprised that mid-Pleistocene hominins were capable and routinely went to quarries and removed materials to nearby safe places for knapping. I would be surprised indeed if this were NOT the case.

See the comment above on our perceived difference of the reviewer's opinion and the prevailing published opinion in the field, as well as the question on cognition; we agree that direct provision does not equal the discovery of a new cognitive threshold, and we haven't argued this here (it is more a reflection of increasing capacities, whenever they exactly emerged though certainly early in the history of our species and earlier as many would have assumed in the past).

For the other part of this comment, this may be a misunderstanding of our main point: at Jojosi, people did not really move the materials far (or to safety) - and this is not part of our main argument - but knapped in more or less on spot, either at the primary outcrop or more frequently within the dongas where large secondary blocks eroded from the primary sources. See our comments further below on what the main argument and key findings from Jojosi are, in our view.

Regarding Twin Rivers, based on Barham (2002) this assemblage is more of a case of longer-distance (>20 km) movement of material and aggregation (such as pigments), and this is also a relevant behavior concerning the increasing range of human-material interactions in the Middle Pleistocene, but not the same behaviors as what we claim for Jojosi (some would call this behavior at Twin Rivers “place provisioning”). There is no evidence or discussion of direct procurement at this site based on the published literature. Interestingly, long-distance movement of materials is much more ubiquitous in the Middle and Late Pleistocene MSA, but regularly associated with/explained by embedded procurement (see references in the Introduction section). We did, however, include Twin Rivers and its evidence in the final paragraph of our Discussion when we discuss the deep roots of increasing human-material interactions as a separate point (Lines 405-406)

Reading the paper, I was led to think of other examples of sites with similarities to Jojosi. While I admit I cannot myself think of site in such a good state of preservation, I think the authors need to include and consider the relative importance of the sites of Gademotta and Kulkuletti (279 ka) in Ethiopia, which are older than Jojosi, and show the specific and repeated exploitation of Obsidian from a source that is further away than the Jojosi source is from the sites there. While at Gademotta other materials are represented in the assemblages, over 94% of all lithics (including debris) are local obsidian, according to Sahle (2013). Of course, Gademotta includes finished artefacts and lacks the spatial and chronological preservation of Jojosi, but I would find it remarkable that the authors do not consider Gademotta as an example of special lithic provisioning, and also likely/probably directly acquisition, as they argue for Jojosi. Gademotta then, at least, needs inclusion as a reference point and comparison regarding its potential direct provisioning and cognitive importance, potentially to illustrate the special importance of Jojosi.

We thank the reviewer for pointing out potential similarities between Jojosi and the important early MSA sites of Kulkuletti and Gademotta. Importantly, our argument is not about long-distance movement of raw material or raw material transport distance; at Jojosi those are a few meters at maximum to the flaking sites (though the produced blanks were exported to other sites somewhere beyond the studied landscape). See the comment below for the main points we are actually arguing for, and hopefully make our main findings clearer. Also, long-distance movement of raw materials can also be (and indeed often is) explained by embedded procurement and movement as personal gear or exchange between groups (see our references in the Introduction).

We re-read the key literature on Gademotta and Kulkuletti (Wendorf & Schild 1974; Sahle et al. 2013; Sahle et al. 2014; Shackley & Sahle 2017; Sahle 2023) and how these sites may be comparable to Jojosi. Only one publication (Sahle & Shackley 2017) assesses this topic more directly. They state on the origin of the non-local obsidian: “Whether this was the result of direct procurement or trade/exchange remains difficult to elucidate”, which supports our main points below on the differences to Jojosi and the problems of differentiating direct/embedded procurement at such sites, which require special, highly-resolved datasets. In general, we noted a similarity in the open-air contexts of these localities and a relatively close-by, high-quality obsidian source (2 km away). That being said, Gademotta and Kulkuletti differ from Jojosi in having a stronger palimpsest character, often in colluvial sediments; they are laterally expansive and also feature other raw materials. Gademotta and Kulkuletti show mostly complete reduction sequences with pronounced technological

variability, abundant tools, retouch debris, and used artefacts (including used projectiles). Accordingly, the interpretation of these sites then tends to favor intense, multi-purpose, long-term occupations instead of a special-purpose, short-term function as raw material provisioning and specialized workshops sites, as for Jojosi. Further, Sahle et al. 2013b note “Gademotta must have provided suitable conditions that supported the relatively continuous, or at least repeated, occupations”, which is much more indicative of residential site activities or a base-camp. Other sites at Kulkuletti are interpreted as large-scale lithic workshops, but again with full reduction sequences (Sahle 2023). According to Shackley & Sahle (2017), some of the obsidian was also sourced from further away, showing multiple and not just one source for the obsidian, which is again different from Jojosi. Based on the published literature, we conclude that Jojosi differs strongly in its content and character from the sites, and the time-averaged nature, site function, and artefact composition of Gademotta and Kulkuletti make it difficult or impossible to distinguish between direct/embedded procurement of the obsidian. Because of the special open-air character and proximity of these sites to a raw material source, we added a new half-sentence specifically relating to these sites in references in lines 316-320 in the context of similar instances of other important MSA rockshelter/cave localities, where it is difficult to demonstrate these activities based on the available archaeological record, even though they might have happened. We also added other citations on these sites in the context of the difference of the specific workshop character to Jojosi.

We thank the reviewer as we think that the comparison with these and other sites helps to clarify our argument about how far Jojosi is a special and best-case scenario for showing specialized provisioning within the Middle Pleistocene record. This argument is also based on additional contextual arguments, such as that apparently no other resources of the landscapes were used, and the visits were very brief and not long-term occupations. A comparison to potential earlier behaviors and precursors in the ESA is now also added with special reference to Wonderboom (see lines 365-377 and comment by reviewer #4).

Furthermore, I think the way that the paper is written, it takes some work on the part of the reader to tease out exactly what the unique situation is at Jojosi. It is not directly stated, and so it takes some time to figure out that it is not the distance from source, and it is not that Jojosi itself is an actual quarry....these were my first assumptions and were never directly countered. The authors could better clarify upfront that Jojosi is a secondary location used for the decortication/early stages of lithic production, and seems to have been exclusively used for this purpose.

The reviewer makes a valid and key point that we should more clearly spell out what the special site function at Jojosi really is and what the basic arguments are for inferring direct procurement; indeed, we think that several of the comments above could likely be answered by this. First, demonstrating direct procurement (or differentiating it from embedded procurement) at a specific site needs several elements: i) a (close-by) raw material source that was the specific target in the past; ii) evidence for its intense and repeated use; iii) an associated workshop, ideally with the main purpose of basic flaking production and the export of elements (otherwise it would be a basic production site and would include potential other tool-assisted activities in the direct surroundings); iv.) exclusion of other activities that were carried out during raw material procurement and flaking (as this would then rather indicate embedded procurement). At Jojosi, we have the unusual resolution and confluence of factors to show all of these points, which makes direct procurement the most probable and most parsimonious explanation. Additional contextual arguments at Jojosi are other available raw materials than hornfels, but its exclusive use, which further supports the role of Jojosi as a locality for specialized raw procurement of one tool stone.

Based on all available evidence presented, the Jojosi site presents a combination of direct raw material procurement localities and specialized workshops (decortication and blank

production plus export without tool production as in typical workshops known from other MSA sites and before!) directly next to a large hornfels primary outcrop and secondary blocks of hornfels eroding from these sources within the dongas. As such, it differs from traditional quarries, as Jojosi 1-7 demonstrate that large blocks from within the dongas were primarily reduced (which did not have to be carried anywhere). That being said, the primary outcrop also has (unstratified) evidence of MSA flaking. Jojosi differs from classic workshop sites in its truncated reduction sequences and a lack of tool production; instead, usable blanks were removed from the site for future use somewhere else, not within the Jojosi landscape (based on both the stratified and surface material), which is a key part of the argument for direct procurement. On top, even though the landscape offers many different resources, no other activities or long-term occupations are evident, which underscores this special, single-purpose function of the presented Jojosi sites.

In regard to changes to the manuscript, we now clarify upfront (at the end of the Introduction section) what exactly is special about the Jojosi sites (lines 91-94) as the reviewer asked. We also more clearly describe and define this situation throughout the text (e.g., at the end of the Results in lines 300-303) and in the Discussion (Lines 337-342; 352-355), also within relation to other sites (i.e., Lotter et al. 2024 for Wonderboom or Gademotta/Kulkuletti; Lines 365-377).

All this draws attention to the need for greater clarity over the theoretical arguments being made here by the authors. Refining these arguments will help demonstrate the special contribution of Jojosi to cognitive debates. Overall, as stated, I think this paper is important, but I do think it would benefit from a more critical eye on the part of the authors concerning their theoretical contextualisation of the sites, and their consideration of direct provisioning as an exciting and cognitively significant finding in the mid-Pleistocene.

Based on the replies and adjustments made in response to the reviewer's comments above, we think the argument we are making is now much clearer and more intelligible, based on multiple lines of evidence and further contextualization of the sites. Again, cognition is just a minor part of our argument; we rather focus on the technological abilities and the increasing flexibility of interacting with the material world (see our updated Discussion).

Finally, I think that some more images of the lithics in the main paper (individual pieces and not only refits as at present) would be interesting and are warranted in light of the importance being claimed for the site. Let us see some technological features rather than just the refits, please.

The reviewer makes a valid point in asking for better images of the lithics in the main paper. We added a new figure of technologically relevant pieces to the Main Text (Figure 5).

Additional points regarding the lithic use-wear analysis.

The description of 'Ad-hoc experiments' for the use-wear does not sound structured and brings uncertainty to the results described. I am left wondering if the work was done in a serious manner. It is lucky then that the positive use-wear identifications were so few, as otherwise I think this expression would be grounds to throw out any results presented. Please amend the description and ensure future studies are correctly and adequately structured.

Line 632 "additional comparisons were made with non-hornfels materials". Greater specificity is needed on the number and type of comparisons drawn – what were the uses of the wider experimental collection used as a reference point?

We thank the reviewer for pointing out the lack of information on the systematic nature of the use-wear analysis and associated experimental work in our initial submission. As a result,

we have added significant amounts of information to the Main methods section (Lines 698-725) and Supplementary Information (Supplementary Information Figures S49 & S50 and Table S46) to show that the study was done systematically and adheres to modern standards in the field. We improved the method section in the main text by adding more details about the experimental protocol and reference collection. We included two new plates in the Supplement: one showing a selection of use traces from the collection and the other comparing pre- and post-use on two experimental flakes. The experiments were conducted systematically, following the case study and controlling variables listed in the new table in the Supplementary Material. We also discuss the wider experimental collections at Tübingen and how far hornfels differs from non-hornfels materials.

Minor but important issues to be addressed:

Line 42: the authors give a date range of 3.0 – 2.6 Myra for the earliest known artefacts. Lomekwi 3 (Harmand et al 2015, as referenced by the authors to substantiate the lower end of this date range), is well dated to 3.3 Ma. 300,000 years is a significant chronological error: if applied to Jojosi we could equally say the site is 0.0 Ma! The stated date range should therefore be amended to 3.3 – 2.6 Myra.

The reviewer is, of course, correct with their suggestion, and we changed this to 3.3-2.6 Mya accordingly.

Please consider amending use of the word *débitage*, when what is meant is debris, or more specifically knapping shatter and fragments. Ditto use of the word 'micro *debitage*'. *Débitage* used as a noun to refer to such material is an inaccurate Anglicisation, even if the misuse of the word is now widespread.

We followed the suggestions and avoided the word *debitage* and instead used more accurate wording for different instances (e.g., micro debris or flaking products).

SI 3, Text S2: the meaning of the word 'transects' is not very clear. I understand the rest of the text, regarding the differentiation of contexts and spits etc, but are these transects are spatial delineation? If possible an illustration of what is meant could be useful here, to show the system of excavation. If not, then for sure greater clarity of explanation is required here as the use of several words with potentially overlapping or indistinct meaning creates some confusion. Adendum: the transects are shown in a photo later in the supplementary information, and this photo or similar could be moved or used to illustrate what is meant by transect on first use.

We thank the reviewer for the detailed and critical reading of our Supplement. We now more clearly explain in SI 3 Text S2 what we mean by transects (spatial excavation units) and refer more clearly to Figure S13, which shows them in the field (plus added the word "transect" to the Figure caption, which was missing previously).

Figures S1-S6: For all the widest-view images, it is important to clarify the scale, either within the images next to the measure seen in the image, or in the figure captions.

The reviewer is correct in pointing out this omission. We now reference scales in the figure captions.

Figures S9 and S10. I don't find the combination of x-axis for depth and y for chronology

clear at all. It would be much better to show both depth and chronology on the same axis, which would allow readers to grasp the date range for each sample, and how these relate to the position of the samples and the depth range of artefacts at each locality. At the present time, it looks like artefacts might be spread throughout MIS5-8 at each locality, which is erroneous and confusing. Figure S10 is similar and needs to be checked for logic and clarity – although noted in the caption, the use of grey for shading makes an implied confusion with Figure S9 and indicates, contrary to the descriptions of the sequences, that artefacts are present at the Triple Junction.

We thank the reviewer for this assessment, and agree that the grey horizontal bars for artefacts in Figure S9 were not ideal and could lead to confusion. Otherwise, we differ in our opinion on the visualization and clarity of Figures S9 and S10. The way the x- and y-axis are shown is standard for age-depth models in chronometric dating, such as luminescence. We also would like to point out that Reviewer #2 finds “Figure. S9 particularly nice” and even considers that we should move it to the main manuscript.

As a response to this reviewer and the positive comments by reviewer #2, we completely remade Figures S9 and S10 and combined them with the previous manuscript Figure 2 into a new, more comprehensive figure providing an overview of the stratigraphic situation and chronometric dating of the Jojosi sites (new Figure 2). We hope that the new visualization and coloring (e.g., removal of grey shadings for artefacts, use of colors for different sedimentary units, same red symbols for artefacts as in the top of Figure 2 to indicate the direct relationships of the artefacts to the ages) clarifies the confusion of reviewer #1 concerning this figure and corresponds to the wishes of a more prominent place of these figures in the manuscript, as reviewer #2 suggested. The previous Figures S9 and S10 are removed from the Supplement.

Figure S19: I do not see the hammerstone being referred to in the caption anywhere, even in the top left as described. Please highlight or reconsider description. Addendum: I later see the hammerstone marked in Figure S21, but it does not immediately appear as a hammerstone from the 2D visual, so I suggest in both images this artefact is highlighted somehow if the authors want to draw attention to the piece. In S19 it is not located on the top left as described.

Figure S20: same concentration but a different orientation and the hammerstone is still said to be in the top left of the image but no hammerstone is clearly identifiable. Addendum: I later see the hammerstone marked in Figure S21, but it does not immediately appear as a hammerstone from the 2D visual, so I suggest in both images this artefact is highlighted somehow if the authors want to draw attention to the piece

The reviewer correctly identifies that the Figure caption of S19 is incorrect. It should read “bottom left”. We also added a red circle and adjusted the figure captions of both Figure S19 and Figure S20 to highlight the position of the hammerstone. There is no hammerstone in Figure S21 and no reference to it, so this was likely a confusion of Figure numbers with Figure S23 (here the hammerstone is marked by an orange dot) by the reviewer.

Reviewer #2 (Remarks to the Author):

Below I evaluate the sedimentological interpretations, luminescence sampling, and luminescence dating, and I suggest elaboration on three points:

- Completeness of the studied archives
- Sampling strategy at Jojosi 7
- Water content assumption

Sedimentological interpretations. The sedimentological interpretations, which provide context for the dating, appear robust and are supported by ample field documentation. One thing I miss is a discussion of how complete the archive is. Specifically, the authors mention that the geologic character prohibited standard trench-style excavation of levels. The four dated sites were identified from outcrop exposures (which makes sense in this environment). Can you comment on if there are many more unidentified artifact lenses likely buried in the sedimentary deposits? If so, what might this add to the interpretation of use of the site?

We appreciate this apt question by the reviewer. Indeed, we reported the results from our systematic survey of the Jojosi landscape in another paper (Will et al. 2024), which reports on a total of 11 identified occurrences, likely *in situ*, with outcropping of large and small non-weathered, hornfels artefacts with fresh edges. By now, we have excavated a total of 4 of them. In the field, the unexcavated artefact concentrations appear in their nature and content similar to Jojosi 1, 5, 6, and 7, and the previous excavations of Jojosi 2-4 (though these are not documented well enough for scientific contextualization). All of these concentrations appear in Unit 4 and at different heights relative to the underlying saprolite (Unit 1). Combined with the information on the high frequency and wide distribution of surface finds from the Middle Stone Age that match the excavated artefacts (more thoroughly reported in Will et al. 2024 and the Supplement), this suggests that these artifact lenses were much more numerous in the past and today are mostly eroded. These observations further underscore our interpretation of the Jojosi landscape as: a) being used repeatedly and over long times b) used for similar raw material procurement and flake reduction as seen in the excavated samples. We now make a clearer point in our manuscript about how this contextual information further supports our main interpretations presented in this paper at the end of the section Site formation and chronology (Line 247-255). We also refer to this information for another comment by reviewer #3 below.

Sampling strategy. The luminescence sampling strategy is appropriate and the results are stratigraphically consistent and convincing. Figure. S9 is particularly nice; you could consider moving this to the main text, possibly with some additional annotation for geomorphic processes (e.g., erosion, deposition) or paleoenvironment. Can you please comment on why an upper sample (to bracket the time of tool production) is not reported at Jojosi 7? Maybe it is too close to the soil?

The reviewer rightly identifies a lack of the upper bracketing of Jojosi 7. Unfortunately, the sedimentary situation did not allow for taking an adequate luminescence sample above the lens due to soil formation processes. We describe this situation in more detail now in Supplementary Information 2, which now reads:

“In the case of Jojosi 7, it was not possible to bracket the artefact layer with luminescence samples due to the intense soil development. We thus only sampled below the artefact layer.”

Luminescence dating. The luminescence dating was performed using a small aliquot pIRIR feldspar approach, which is appropriate for this setting and timeframe. The protocol was previously tested and published, which adds to the robustness. Analytical details and test results are well reported in the supplement. I agree with the decision not to do a fading correction. One aspect of the luminescence dating that could be better justified is the water content assumption. In situ values yielded 15% for all samples, and 5% (or ~33% relative) uncertainty was used. This is a healthy uncertainty, but I miss a discussion of whether 15% is a reasonable value for long term water content in this environment (i.e., going back to the Pleistocene). Is there seasonality in water content here and, if so, were samples collected under wet, dry, or intermediate seasonal conditions? Has the wetness of the environment changed significantly over the last several hundred thousand years? Please add some (paleo)environmental information to support the use of the in situ values.

Thank you very much for your question. Based on today's seasonality, our time of sampling, as well as modelled past precipitation variability by Krapp et al. (2021), we reason that the suggested water content and uncertainties are representative for the time period dated. To clarify this for the reader, we modified the text in the Supplementary Material 2, which now reads:

"We reason that this water content represents probable scenarios over the dated age range. Firstly, we collected the samples in autumn and rain events had taken place prior to our field campaign, likely wetting the sediments. Krapp et al. (2021) showed that the mean annual temperature only varied between 5 and 10 % for our time period of interest. Furthermore, our luminescence dating results indicate that similar environmental conditions resulting in cut-and fill processes and the donga landscape, must have been present over at least the past ~600 ka, resulting in well-drained dongas."

Specific comments:

L. 34. "On" -> "Of"?

Changed the phrasing here

L. 47. "But" -> "And"

Changed accordingly.

L. 159. What does "low skeleton content" mean? This is not a sedimentological term I am familiar with. If this is a clastic attribute, you might consider rephrasing to avoid the (archaeological) connotation of (human) skeletons.

The term "skeleton content" is used in soil science to describe coarse clasts >2 mm, insinuating its supporting role in providing structure and stability to the soil. Due to the proximity of artifacts and soil formation, we have decided to address this profile with a soil description, complementary to the sedimentological approach. We understand that this term might be misleading in an archaeological context, and therefore changed the wording to "coarse clast content".

L. 163-172. This would fit better in the Methodology section.

While we agree with the reviewer that this is a methods statement, we feel that the information is important to understand key aspects of the field approach and recovery that are directly relevant to the presentation of results (e.g., the documentation of highly resolved archaeological contexts) and further interpretation; as such, we decided to keep this here.

L. 221-223. This is also a methods statement. Better to dive right in to presenting luminescence dating results.

We rewrote this (introductory) section to better fit in the results section and remove parts of the methods statement.

L. 310. Perhaps change "sole" to "primary", because it is not possible to truly know all the motivations of prehistoric people, and it is possible that some additional (secondary?), unidentified benefits were obtained from visiting the sites (e.g., of sociocultural value like mentioned in L. 346).

We agree with the reviewer and removed "sole" and replaced it with less absolute qualifiers throughout the manuscript (a total of 3 instances).

L. 570. Here or somewhere in the main-text methods section, it would be good to state where the luminescence prep/analyses were performed (i.e., Cologne Luminescence Laboratory; University of Cologne).

We added the following text to the method section of the main text (Lines 631-634)

Sample preparation for equivalent dose and for dose rate determination were conducted in the Cologne Luminescence Laboratory (CLL) at the University of Cologne. Luminescence measurements were carried out at the CLL and at Risø (Technical University of Denmark, DTU).

L. 582-591. Here it might be useful to refer to Fig. 2.

We refer to Figure 2 now in each of these paragraphs.

Reviewer #3 (Remarks to the Author):

I have no major criticisms of the manuscript or the methods, the suggestions below are offered to clarify some minor items:

-The landscape is described as resource rich (Ln 129 and Ln 316), but this seems to be an assessment primarily of conditions in the present. The passages and supplement reference a stream as well as the lithic source, but while the latter was evidently accessible in the past it's not clear to me what condition the stream was in, or any other aspect of the environment for that matter, during the timeframes of interest. I find a mixed bag when I look at paleoclimate reconstructions from the region, with Toffolo et al (2017) suggesting a dry phase around 240-200k at nearby Florisbad. I suggest adding a reference or two to better justify the claim of resource richness in the past.

Our main argument here was that the current and (likely) past situation of Jojosi is one that presents various resources relevant to hunter-gatherers: Diverse raw materials (hornfels, dolerite - primary outcrops; quartz, quartzite pebbles in the river), drinking water (waterfall, river), shelter (rockshelter close by), and edible plants and game. The interesting observation is then that during the MSA, people used this landscape primarily as hornfels procurement and specialized workshop sites, ignoring most other aspects of the available resources, which further strengthens our argument for purposeful, direct procurement. As a first reply, we changed "resource richness" to "resource diversity" as this is really what we wanted to convey for Jojosi; secondly, we added relevant references (see below) that suggest a mostly similar climate and environment for this region in the studied period.

In the absence of the preservation of local paleoenvironmental proxies (other than the zooarchaeology, which indicates that game was principally indeed around), we are also currently conducting paleoclimate modelling that is not yet published, though the preliminary findings fit generalized reconstructions for the wider area (e.g., Partridge et al. 1998). The fact that the occupations span several cycles of orbital precession suggests variation in summer insolation, and consequently, variation in both total precipitation and precipitation seasonality. The paleo-precipitation reconstruction from the Tswaing dataset, going back up to 200 ka BP (Partridge et al., 1998), indicates generally higher-than-modern rainfall during the Jojosi 5 and Jojosi 1 occupations. The global paleoclimate reconstruction by Krapp et al. (2021) suggests precipitation levels 5–10% higher than modern for Jojosi 1 and Jojosi 6, with a similar upward tendency for Jojosi 7. In contrast, conditions for Jojosi 5 are more uncertain, with model outputs indicating both higher- and lower-than-modern rainfall for the occupation period. Their seasonality reconstruction gives mixed results, indicating occupations with similar to modern rain season conditions, but also occupations with longer or shorter rain seasons. We therefore agree, that the climate signals are inconclusive, but given the orographic effect due to the small escarpment and nearby peaks like Mount

Telezini, an yet undated river terrace, and the observation of hydromorphic markers in the sediments above modern channel bed, that hydrometeorologic conditions allowed for at least seasonal flow of Jojosi River and its positive effect on the area's flora and fauna. More generally, our simulations show that the area shifted between Grassland and Savannah biomes, which both offer general plant and animal resources to past hunter-gatherers.

We added the following text to the manuscript in direct response to the reviewer's comments in Lines 130-138:

“Our geological surveys documented a landscape with diverse resources, including drinking water from the adjacent river and a nearby waterfall, various accessible rock resources and several cliffs that provide shade and shelter (Figure 1). Situated in a topographically diverse landscape with orographic effects on local climate, the area offers various plant and animal resources at the grassland–savanna interface³². Palaeoclimatic reconstructions indicate fluctuating conditions for the general area that shifted between grassland and savanna environments, with precipitation levels and seasonality sometimes slightly lower and sometimes higher than today during the last 200,000 years, providing enough rainfall to support at least a seasonal flow of the Jojosi River^{43,44}. “

We also adjusted our statement in the Discussion in Lines 345-346.

-The lithic accumulations at Jojosi are characterized as being generated over “tens of thousands of years” (ln 31, 92, 233), suggesting a long term accumulation and/or frequent visitation. Based on the dating (ln 560-591), it seems clear that the site was visited more than once over such a window, but the number of separate visits is not fully clear given overlaps in the date ranges. The Jojosi-1 dates, at 1 sd, provide the best evidence for an extended period of accumulation, but actual duration over which any of these assemblages formed within their accumulation windows is necessarily ambiguous. While I believe the authors are probably correct in their interpretation, I suggest rephrasing some of these passages to account for alternative interpretations of the chronology.

We understand that some further clarification of the overall frequency and duration of the visits to the entire Jojosi landscape is needed. The key for us are the bracketed ages of Jojosi 1, 5 and 6 (Jojosi 7 provides only a maximum age and is consistent with Jojosi 6): At 1 sigma and incorporating stratigraphic information, Jojosi 6 dates most likely around ~220 ka (full bracket 201-258 ka), Jojosi 5 at ~160 ka (full bracket 136-187 ka) and Jojosi 1 at ~110 ka (full bracket 106-139 ka). Considering that Jojosi 5 and 6 do not overlap, we have at least 2 phases of site use. There is minimal overlap between Jojosi 1 and 5, but only in the underlying vs. overlying date that brackets the occurrence, so the most likely interpretation is again different times of usage. In addition, there is a marked technology difference at Jojosi 1, which is the only site that features clear use of Levallois, which adds a contextual argument to being a different (later) phase of use. Finally, the location of the artefact lenses in Unit 4 in relation to Unit 1 differs markedly (some directly on top, others further away), again providing a relative sense of longer duration and episodic visits to the landscape.

Importantly, these ages and further considerations for the four excavated sites give us only the absolute minimal potential duration and frequency of occupations at Jojosi: our systematic survey of the Jojosi landscape (Will et al. 2024) reported on a total of 11 identified comparable occurrences, and this is likely an incomplete identification of what is still there (plus most stratified evidence has been eroded). Combined with the information on the high frequency and wide distribution of surface finds from the Middle Stone Age that match the excavated artefacts (more thoroughly reported in Will et al. 2024 and the Supplement), artifact lenses must have been much more numerous in the past and are today mostly eroded. These observations further underscore our interpretation of the Jojosi

landscape as: a) being used repeatedly in short episodes and over long times (multiple tens of thousands of years) and b) used for similar raw material procurement and flake reduction as seen in the excavated samples. Based on these arguments, we consider our chronological interpretation of the overall duration of the use of the Jojosi landscapes (tens of thousands of years between ca 220-110 ka) to remain conservative in our assessment of the true duration of the occupations within the landscape.

In combination with the reply to reviewer #2 above on the occurrence of other (non-excavated) stratified artefact lenses and the arguments provided here, we sharpen and clarify our chronological interpretation of the overall interpretation within the main Manuscript text based on all these different lines of evidence (lines 239-255).

The luminescence ages do not help us to resolve the actual duration over which any of these assemblages formed. Based on contextual (refits!) and sedimentological arguments, however, they appear to be very short, possibly only a few hours: The artefact lenses are isolated occurrences in their sediment body vertically and horizontally, the sediment built-up was quite fast, and there are extensive refits indicating contemporaneity for at least many of the artefacts. In contrast, we have no indications for longer-term site use or palimpsests. We also sharpened our main manuscript text to reflect these arguments.

-Table S8 is missing its summed totals. Also, I'll admit the densities of 2,000,000 artifacts per cubic meter raised my eyebrow. Extrapolating values from 0.003 m³ excavated to a 1 m³ comparative standard seems like a less-than-intuitive measure, though I'm not exactly sure how to handle this for comparisons sake. I understand that a smaller reference volume is not conventional, but perhaps something like dm³ would be more appropriate. Alternatively, since the table caption already mentions relative orders of magnitude between overburden and artifact lenses, I wonder if it might be more useful here to express these densities as multiples relative to the respective overburdens (e.g, Jojosi-6 Lens 2 = 1433 x overburden)?

We thank the reviewer for their detailed read of our Supplement! We added the missing summed totals to Table S8. We used the m³ comparison for density mainly as it is a measure that also exists for other sites, and as a measure for comparison across the different contexts. We agree that the extrapolation from necessarily very small values to cubic meters leads to potentially less intuitive numbers. We like the idea of expressing the density of material of the lenses as multiples relative to the respective overburdens and added it to Table S8 and mentioned it in the Main text as an alternative measure (lines 183-185)

Reviewer #4 (Remarks to the Author):

I found the paper to be well-written and convincing with good use of available data. I have no major issues with the manuscript but I have one comment. It would interesting to see how the Jojosi sites relate to other lithic workshops and flake-harvesting localities in southern Africa. I'm thinking particularly of Wonderboom near Pretoria in South Africa. Although that is an Acheulean site, I think it could be of interest to this project in understanding what Will et al call 'tool stone' provisioning. Like Jojosi, it too had little evidence of fauna suggesting it was a specific lithic workshop, and, similar to Jojosi, the focus of the workshop was specifically on one type of material – in that case, quartzite. Indeed, the fact that flake and lithic resource extraction sites and workshops occurred in this region in the Mid Pleistocene (preceding modern Homo sapiens) would add an interesting dimension to what is found at the Jojosi sites.

We thank the reviewer for bringing up the highly relevant site of Wonderboom and for asking us to contextualize our findings within the broader Middle Pleistocene (late ESA) record. We

now include both a concrete comparison with the evidence at Wonderboom and discuss our findings more broadly in the context of the Middle Pleistocene ESA record in Lines 365-377 and Lines 383-385. While we concede that some precursor behaviors may be recognizable in the ESA, we do not know of any instance of a demonstrated direct procurement, which also has much to do with the lower resolution of the older record (e.g., many open-air sites; few dated occurrences; difficulty to show that sites were only use for raw material extraction and no other activities). In the specific case of Wonderboom, while the flake-harvesting from bedrock and secondary blocks is indeed reminiscent of Jojosi - here we also have flaking on the outcrops and full reduction of large blocks from within the dongas - there are some important differences: Wonderboom has evidence for full reduction sequences and the production of end products such as LCTs, and other activities that might have been carried out here cannot be precluded with confidence based on the complex site formation processes. An interesting and diverging aspect of Jojosi from Wonderboom and other Acheulean workshop sites is a truncated reduction sequence and the absence of intense tool production (at both the stratified and surface sites); which again underlines the different and special character of these sites. As a result, we think that by integrating these sites and comparing them to Jojosi, we could further sharpen our main arguments for direct provisioning.

Minor edits:

Ln 45. Change 'central' to key'.

Changed accordingly

Ln 54-59: Include a few references for these modes.

Here we refer directly to the key references 1-3 from the previous sentence, and considering the limit of references in Nature Communications refrained from adding more (though we are aware that they exist).

Ln 338. Replace 'well into' with 'from'.

Changed accordingly.

REPLY TO REVIEWERS

Reviewer #1 (Remarks to the Author):

I have read the revised manuscripts provided by Will et al. and the replies to the reviewers. The authors have done a detailed and balanced job in addressing my concerns and those of the other reviewers. I am particularly pleased that they were able to clarify better the precise behavioural signature they have identified at Jojosi and their argument for its wider evolutionary significance, as well as changes to some of the figures in the main paper and to the description of the use-wear analysis, which now appears as a well-developed and professionally conducted part of the overall study.

I think that the results of this study are noteworthy, and the claims are supported by the data presented. I recommend this manuscript for publication and pass my congratulations to the research team.

We thank the reviewer for the praise of our article, and that the changes we made were sufficient for the previous concerns. As a result, we did not further modify the content of our article.

“Specialised and persistent raw material procurement by humans in the Middle Pleistocene”

Review for *Nature Communications* by Liz Chamberlain

The manuscript by M. Will and colleagues makes a case for intentional and direct acquisition of lithic raw materials by Pleistocene-aged humans. The authors state this finding is contradictory to the currently accepted view that people of this time indirectly gathered lithics while conducting other, priority tasks. I was asked to review the geochronology aspects of this manuscript. I will not comment on the anthropological aspects which are outside my specialization, except to say that I found the authors’ narrative to be plausible and well-presented.

Below I evaluate the sedimentological interpretations, luminescence sampling, and luminescence dating, and I suggest elaboration on three points:

- Completeness of the studied archives
- Sampling strategy at Jojosi 7
- Water content assumption

I also make a few specific comments which may help to finetune the manuscript. If the findings and interpretation hold up under review by anthropologists, this has the potential to be an important article worthy of publication in *Nature Communications*.

Sedimentological interpretations. The sedimentological interpretations, which provide context for the dating, appear robust and are supported by ample field documentation.

One thing I miss is a discussion of how complete the archive is. Specifically, the authors mention that the geologic character prohibited standard trench-style excavation of levels. The four dated sites were identified from outcrop exposures (which makes sense in this environment). Can you comment on if there are many more unidentified artifact lenses likely buried in the sedimentary deposits? If so, what might this add to the interpretation of use of the site?

Sampling strategy. The luminescence sampling strategy is appropriate and the results are stratigraphically consistent and convincing. Figure. S9 is particularly nice; you could consider moving this to the main text, possibly with some additional annotation for geomorphic processes (e.g., erosion, deposition) or paleoenvironment. Can you please comment on why an upper sample (to bracket the time of tool production) is not reported at Jojosi 7? Maybe it is too close to the soil?

Luminescence dating. The luminescence dating was performed using a small aliquot PIRIR feldspar approach, which is appropriate for this setting and timeframe. The protocol was previously tested and published, which adds to the robustness. Analytical details and test results are well reported in the supplement. I agree with the decision not to do a fading correction.

One aspect of the luminescence dating that could be better justified is the water content assumption. In situ values yielded 15% for all samples, and 5% (or ~33% relative) uncertainty was used. This is a healthy uncertainty, but I miss a discussion of whether 15% is a reasonable value for long term water content in this environment (i.e., going back to the Pleistocene). Is there seasonality in water content here and, if so, were samples collected under wet, dry, or intermediate seasonal conditions? Has the wetness of the environment changed significantly over the last several hundred thousand years? Please add some (paleo)environmental information to support the use of the in situ values.

Specific comments:

L. 34. “On” → “Of”?

L. 47. “But” → “And”

L. 159. What does “low skeleton content” mean? This is not a sedimentological term I am familiar with. If this is a clastic attribute, you might consider rephrasing to avoid the (archaeological) connotation of (human) skeletons.

L. 163-172. This would fit better in the Methodology section.

L. 221-223. This is also a methods statement. Better to dive right in to presenting luminescence dating results.

L. 310. Perhaps change “sole” to “primary”, because it is not possible to truly know all the motivations of prehistoric people, and it is possible that some additional (secondary?), unidentified benefits were obtained from visiting the sites (e.g., of sociocultural value like mentioned in L. 346).

L. 570. Here or somewhere in the main-text methods section, it would be good to state where the luminescence prep/analyses were performed (i.e., Cologne Luminescence Laboratory; University of Cologne).

L. 582-591. Here it might be useful to refer to Fig. 2.